# Biologically derived epicardial patch induces macrophage mediated pathophysiologic repair in chronically infarcted swine hearts

J. J. Lancaster[1], A. Grijalva[1], J. Fink[2], J. Ref [1], S. Daugherty[1], S. Whitman[1], K. Fox[1,3], G. Gorman[1], L. D. Lancaster[1], R. Avery[1], T. Acharya[1], A. McArthur[1], J. Strom[1], M. K. Pierce[1], T. Moukabary[1], M. Borgstrom[4], D. Benson[1], M. Mangiola[5], A. C. Pandey [6], M. R. Zile[7], A. Bradshaw[7], J. W. Koevary[1,8] & S. Goldman[1 ✉]

There are nearly 65 million people with chronic heart failure (CHF) globally, with no treatment directed at the pathologic cause of the disease, the loss of functioning cardiomyocytes. We have an allogeneic cardiac patch comprised of cardiomyocytes and human fibroblasts on a bioresorbable matrix. This patch increases blood flow to the damaged heart and improves left ventricular (LV) function in an immune competent rat model of ischemic CHF. After 6 months of treatment in an immune competent Yucatan mini swine ischemic CHF model, this patch restores LV contractility without constrictive physiology, partially reversing maladaptive LV and right ventricular remodeling, increases exercise tolerance, without inducing any cardiac arrhythmias or a change in myocardial oxygen consumption. Digital spatial profiling in mice with patch placement 3 weeks after a myocardial infarction shows that the patch induces a CD45[pos] immune cell response that results in an infiltration of dendritic cells and macrophages with high expression of macrophages polarization to the anti-inflammatory reparative M2 phenotype. Leveraging the host native immune system allows for the potential use of immunomodulatory therapies for treatment of chronic inflammatory diseases not limited to ischemic CHF.

[1] Sarver Heart Center, Department of Medicine, University of Arizona, 1501 North Campbell Avenue, Tucson, AZ 85724, USA. [2] Division of Blood & Marrow Transplant & Cellular Therapy, Department of Pediatrics, Masonic Cancer Center, University of Minnesota, Minneapolis, MN 55455, USA. [3] Division of Cardiothoracic Surgery, Department of Surgery, University of Arizona, 1501 North Campbell Avenue, Tucson, AZ 85724, USA. [4] Research & Discovery Tech, Research Computing Specialist, Principal, University of Arizona, 1501 North Campbell Avenue, Tucson, AZ 85724, USA. [5] Department of Pathology, NYU Grossman School of Medicine, New York City, NY 11016, USA. [6] Section of Cardiology, Tulane University Heart and Vascular Institute, John W. Deming Department of Medicine, Section of Cardiology, Department of Medicine, Southeast Louisiana Veterans Healthcare System, Tulane University School of Medicine, New Orleans, LA 70122, USA. [7] Ralph H. Johnson VA Medical Center, Division of Cardiology, Medical University of South Carolina, Thurmond/ Gazes Building, 30 Courtenay Drive, Charleston, SC 29425, USA. [8] Biomedical Engineering, College of Engineering, University of Arizona, 1127 E. James E. Rogers Way, Tucson, AZ 85721, USA. ✉email: goldmans@shc.arizona.edu

Chronic heart failure (CHF) affects nearly 65 million people world-wide with a prevalence of 8.5 per 1,000 people, and an expected 46% increase by 2030[1,2]. Chronic heart failure is a leading cause of hospital readmissions and death in the US with ~40–50% mortality rate within 5 years of diagnosis. There are approximately 6 M people with CHF in the US with a nearly $40B annual cost of treatment[1]. In the US, the most common cause of heart failure with reduced ejection fraction (HFrEF) is coronary artery disease that results in ischemic damage classically from a myocardial infarction (MI). While medical management for ischemic CHF can improve symptoms, partially reverse left ventricular (LV) remodeling, improve quality of life, and increase life expectancy, the burden of the disease in terms of increasing hospitalizations and mortality remains mostly unabated[2]. The advances in management for ischemic heart failure have mostly focused on different approaches to neurohormonal blockade. However, ischemic CHF is a progressive disease and currently there is no treatment directed at the pathologic cause of ischemic CHF, the loss of functioning cardiomyocytes.

Despite the promise that regenerative medicine could potentially address this pathology, progress has been limited. The initial reports of transplanting human cells into the infarcted hearts that could replace or produce cardiomyocytes have encountered limitations, that resulted in challenges to the entire field by limiting enthusiasm of investigators and funding agencies to pursue this area of investigation[3,4]. These limitations included that c-kit+ cells minimally contributed cardiomyocytes to the heart and some previous publications contained experimental flaws. Additionally, reports of improvements in LV function in animal models of acute MI (not CHF) stimulated clinical trials in patients with CHF that did not result in beneficial effects largely because the pathology of an acute MI is different than that of CHF[5]. Lastly, although positive outcomes from cell-based therapy are reported, the mechanism(s) of action are not well understood[6–11].

Our prior studies showed improvements in LV function and pathophysiology with an epicardial patch composed of human induced pluripotent stem cell derived cardiomyocytes (hiPSC-CMs) and human neonatal fibroblasts implanted on a bioresorbable matrix in a rat model of ischemic CHF[12]. Notably, the animals were not immune suppressed, the hiPSC-CMs were well integrated into the damaged myocardium and did not persist beyond 30 days after implantation. Thus, as opposed to transplanting human cells into the damaged heart, our approach shows that the transplanted cells do not have to survive to create a positive physiologic benefit; this is a non-integrated approach to cell-based therapy.

In this report, we show this patch implanted on the epicardial surface of immune competent Yucatan mini swine 1 month after MI improved LV pathology, restored LV contractility without any constrictive effect on LV filling, partially reversed maladaptive LV and right ventricular (RV) remodeling, increased exercise tolerance, and caused no ventricular arrhythmia, after 6 months of treatment. Myocardial oxygen consumption and the rate-pressure product did not change indicating no change in myocardial energy requirements.

Based on these positive physiologic benefits in a large animal model and the fact that the cells were rapidly cleared, we hypothesized that the cardiac patch induced changes leading to repair within one month after patch placement. To study mechanisms, we used digital spatial profiling (DSP) to explore mechanisms of action in mice after a MI. Digital spatial profiling utilizes immunostaining, imaging, transcriptomics, and sequencing to allow the selection of specific areas of tissue and provides information about tissue histology, protein expression, and spatial location of RNA transcripts within different areas of the heart, i.e., infarct zone and patch zone in the same animal. The patch induced a CD45[pos] immune cell response that resulted in the infiltration of dendritic cells and macrophages. Further analysis revealed the expression of M2 markers that are associated with the reparative M2 phenotype. This shows the potential for an epicardially placed patch of hiPSC-CMs and NDFs to treat ischemic CHF by coordinating the allograft immune response towards repair.

## Results

**Hemodynamics in mini swine**. To determine the hemodynamic effects of the patch, data were collected at baseline, 1-month post-MI and 6 months post-treatment (Table 1). Body weights increased at 1-month post-MI compared to baseline and at 1-month post-MI compared to the 6month post-treatment controls but are within normal ranges for age and body type of the swine. Right heart data collected using a Swan-Ganz Catheter, showed no changes in pulmonary artery, pulmonary artery wedge or right atrial pressures occurred throughout the study (Table 1).

Conductance catheters were used to collect left heart data, including heart rate and LV pressure/volume data (P/V) data. There were no changes in heart rate, LV pressures or ± LV dP/dt. Tau was significantly prolonged in the 6-month post patch treated group compared to 1month post-MI (Table 1). Notably, the rate pressure product and the calculated $MVO_2$ did not change indicating myocardial energy utilization did not change with patch application. At 1 month after MI, there were increases in LV end-systolic volume (ESD) and LV end-diastolic (ED) volumes with decreases in ejection fraction (EF) (Table 1) and decreases in ES P/V relationship or end-systolic elastance (Ees) and the slope of the LV ED pressure volume (LV-EDPVR) relationship (Table 1, Fig. 1). The changes at 1 month after MI are consistent with other reports of mini swine post-MI models, with increases in LV volumes and maladaptive LV remodeling[13–15].

At 6-months after patch placement, the Ees increased back to the baseline level with the patch but remained decreased in the control swine (Table 1, Fig. 1) showing an increase in LV contractility with the patch. Normalizing Ees for LV mass: volume resulted in similar results, i.e., return to baseline (Table 1). This improvement in LV contractility occurred with no change in the LV-EDPVR with 6 months of treatment indicating that LV systolic function improves, and LV filling is not altered by the patch, an observation supporting evidence that the patch did not result in constrictive physiology (Fig. 1).

**Cardiac magnetic resonance imaging**. Cardiac magnetic resonance (CMR) was performed to define structural changes in the left and right ventricles at baseline, 1-month post MI, 1, 3- and 6-months post treatment (Table 1, Fig. 2). At 1 month after the MI, there were increases in LV ES volume, LV ED volume, RV ED volume, a decrease in EF, and a decrease in LV mass: volume. There was confirmatory evidence of LV dysfunction 1 month after the infarct with LV remodeling based on agreement between CMR and conductance catheter chamber volume data. The average infarct size was 24.8 ± 3.6% of the left ventricle. With the decreases in Ees, EF, and increases in LV volumes, this swine model created similar structural and functional changes characteristic of patients with ischemic CHF. The structural changes in swine and mice are the same changes documented in our previous publications in Sprague-Dawley rats[12,16].

**LV and RV remodeling**. At 6 months after implant, LV ED and ES volumes remained increased in the control swine but showed a decreasing trend after treatment with the patch (Table 1, Fig. 1). Right ventricular volumes increased both at 3 months and 6 months in the control animals, while RV ED volume decreased

**Table 1 Study Parameters.**

| Study Parameters | Baseline (n = 14) | 1-month post-MI (n = 14) | 6-month post-Treatment CHF Control (n = 6) | 6-month post-Treatment Patch (n = 8) |
|---|---|---|---|---|
| Body Weight (kg) | 46.6 ± 1.1 | 47.9 ± 1.1* | 62.3 ± 6.6# | 52.8 ± 3.5 |
| Heart Mass (g) | NA | NA | 233.3 ± 9.9# | 199.5 ± 4.4 |
| Right Heart Parameters | (n = 10) | (n = 11) | (n = 4) | (n = 7) |
| PA Systolic Pressure (mmHg) | 22.0 ± 2.1 | 18.3 ± 1.0 | 19.4 ± 2.9 | 19.5 ± 2.0 |
| PA Diastolic Pressure (mmHg) | 10.3 ± 1.7 | 6.4 ± 1.3 | 8.5 ± 0.8 | 7.5 ± 1.6 |
| PA Wedge Pressure (mmHg) | 7.6 ± 1.4 | 4.8 ± 1.2 | 6.7 ± 0.5 | 6.9 ± 1.8 |
| Cardiac Output (L/min) | 4.1 ± 0.8 | 3.5 ± 0.2 | 3.7 ± 0.2 | 3.3 ± 0.3 |
| Right Atrial Pressure (mmHg) | 4.9 ± 0.8 | 3.3 ± 1.5 (n = 10) | 3.8 ± 0.3 | 3.3 ± 1.0 (n = 6) |
| Conductance Catheter Parameters | (n = 14) | (n = 14) | (n = 6) | (n = 8) |
| Hemodynamics | | | | |
| Heart Rate (bpm) | 85.3 ± 3.5 | 81.8 ± 3.5 | 93.3 ± 7.0 | 82.4 ± 5.5 |
| LV Systolic Pressure (mmHg) | 86.9 ± 2.9 | 86.9 ± 3.8 | 87.7 ± 4.5 | 87.1 ± 3.2 |
| LV End Diastolic Pressure (mmHg) | 10.2 ± 1.1 | 10.0 ± 1.2 | 8.8 ± 2.3 | 7.1 ± 1.6 |
| Tau (ms) | 41.7 ± 2.6 | 41.7 ± 1.7 | 45.0 ± 3.0 | 49.4 ± 3.2# |
| +dP/dt (mmHg/sec) | 1277 ± 61 | 1397 ± 65 | 1355 ± 76 | 1421 ± 95 |
| -dP/dt (mmHg/sec) | -2024 ± 117 | -2104 ± 135 | -1983 ± 108 | -1956 ± 114 |
| MV$_{O_2}$ (µmol*min$^{-1}$*g$^{-1}$) | 1.72 ± 0.06 | 1.70 ± 0.07 | 1.86 ± 0.11 | 1.70 ± 0.08 |
| Rate Pressure Product (mmHg*bpm) | 7389 ± 360 | 7106 ± 425 | 8133 ± 600 | 7112 ± 436 |
| Conductance Catheter Derived Parameters | (n = 11) | (n = 13) | (n = 6) | (n = 7) |
| LV End Systolic Volume (mL) | 28.0 ± 4.3 | 48.5 ± 3.6* | 50.0 ± 12.4 | 42.4 ± 9.6 |
| LV End Diastolic Volume (mL) | 59.1 ± 4.1 | 83.1 ± 4.5* | 91.0 ± 16.9 | 69.4 ± 10.7 |
| LV Ejection Fraction (%) | 57.4 ± 3.6 | 46.6 ± 2.6* | 54.9 ± 4.3 | 48.5 ± 5.6 |
| Ees | 1.85 ± 0.2 | 0.99 ± 0.07* | 1.07 ± 0.23* (n = 5) | 2.1 ± 0.21‡ (n = 8) |
| EDPVR | 0.13 ± 0.01 | 0.11 ± 0.01 | 0.06 ± 0.02#* (n = 5) | 0.14 ± 0.01‡ (n = 8) |
| Ees/LV Mass:Volume | 1.52 ± 0.19 (n = 9) | 1.03 ± 0.07* (n = 11) | 1.09 ± 0.26 (n = 5) | 1.94 ± 0.17‡ (n = 6) |
| Magnetic Resonance Imaging Parameters | (n = 12) | (n = 12) | (n = 6) | (n = 6) |
| LV End Systolic Volume (mL) | 22.05 ± 1.3 | 34.7 ± 3.0* | 36.4 ± 3.3* | 32.9 ± 4.7* |
| LV End Diastolic Volume (mL) | 56.4 ± 2.9 | 71.9 ± 3.8* | 81.8 ± 7.0* | 63.8 ± 6.7 |
| Stroke Volume (mL) | 34.4 ± 2.2 | 38.1 ± 3.5 | 40.0 ± 2.3* | 31.3 ± 2.7‡ |
| LV Ejection Fraction (%) | 60.8 ± 1.6 | 51.7 ± 3.1* | 55.4 ± 2.2 | 49.3 ± 3.5* |
| LV Mass Diastolic (g) | 63.0 ± 1.9 (n = 14) | 76.2 ± 5.1* (n = 14) | 85.7 ± 8.6* (n = 6) | 71.8 ± 5.0 (n = 8) |
| LV Mass:Volume (g/mL) | 1.1 ± 0.1 | 1.0 ± 0.2 | 1.1 ± 0.1 | 1.2 ± 0.1 |
| LV Infarct Size (%) | 0 ± 0 | 0 ± 0 | 23.5 ± 4.7 (n = 6) | 24.9 ± 4.9 (n = 8) |
| RV End Systolic Volume (mL) | 13.9 ± 1.7 | 15.4 ± 2.2 | 19.5 ± 3.9 | 15.7 ± 3.1 |
| RV End Diastolic Volume (mL) | 44.3 ± 3.2 | 49.4 ± 4.7 | 61.7 ± 7.1*# | 45.9 ± 2.5‡ |
| RV Ejection Fraction (%) | 67.4 ± 3.9 | 68.0 ± 4.3 | 68.7 ± 4.5 | 65.9 ± 6.2 |
| T2 Mean Values (ms) | 54.6 ± 2.4 (n = 7) | 77.6 ± 5.3* (n = 9) | 77.6 ± 8.6* (n = 6) | 53.9 ± 2.0#‡ (n = 3) |

*MI* myocardial infarction, *LV* left ventricle, *PA* pulmonary artery.
Data presented as Mean + SEM.
*$P < 0.05$ vs Baseline; #$P < 0.05$ vs 1mo pMI; ‡$P < 0.05$ 6mo pTX CHF Control

both at 3 months and 6 months after patch placement (Fig. 2). The data for LV volumes at 3 months was the same as at 6 months. Thus, application of the patch alters this progression and reduces RV diastolic volume demonstrating organ level structural improvements. The decrease in LV diastolic mass and total heart mass with treatment is also evidence of an improvement in organ level function with the patch (Table 1).

**T2 MRI relaxation.** The MRI derived Global myocardial T2 relaxation time decreased with the cardiac patch at 6 months indicating a decrease in myocardial edema (Table 1).

**Arrhythmia monitoring and activity levels.** The Internal Cardiac Monitor (ICM) recordings of continuous ECGs 24/7 showed there were no ventricular arrhythmias in either the control or patch treated swine. Interestingly, these Yucatan mini swine had

wide variations in their sinus rates during activity, ranging from sinus bradycardia in the 40 beats/min while asleep to sinus tachycardia of over 200 beats/min when active. The patch treated swine showed a trend of increasing their total overall activity levels after treatment, while all swine were able to complete similar levels of treadmill exercise with appropriate heart rates during exercise and recovery (Fig. 3). Data collected using Fit-Bark® activity monitors, worn on collars for the duration of the study, showed a trend toward increasing treadmill and overall exercise activity after treatment with the patch (Fig. 4) with similar increases in heart rate with exertion for both groups. The observation that heart rate was not different in the patch treated group with exercise is further suggestive evidence that the patch did not create constrictive physiology.

**Pathology & histopathology.** There was one swine death during the MI procedure due to intractable ventricular fibrillation; there

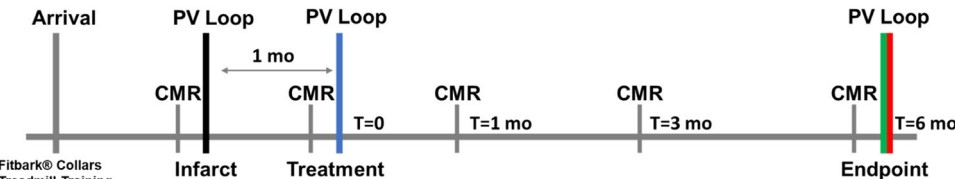

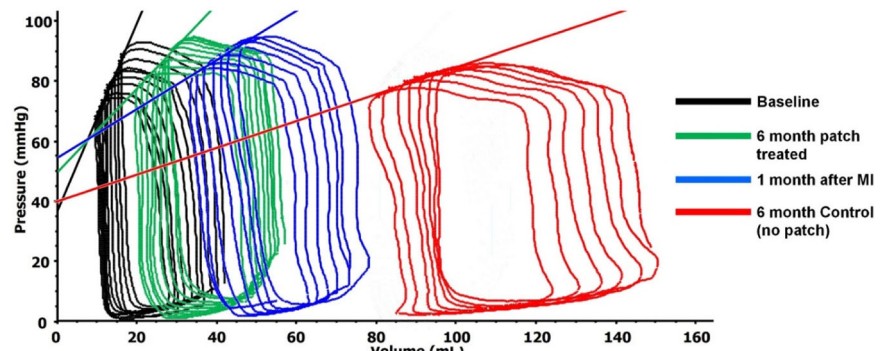

**Fig. 1 Chronologic swine study timeline and Pressure/Volume relationships.** The myocardial infarction was performed by cardiac catheterization occlusion/reperfusion, 4 weeks prior to thoracotomy to place patch with cells or inert-matrix placement. The swine underwent CMR imaging, hemodynamic measurements, conductance catheter pressure/volume assessments, activity monitoring, treadmill testing, ECG loop recorders 24/7 and histopathologic examination at necropsy. CMR; cardiac magnetic resonance. Conductance catheter composite LV P/V analyses showed a decrease in the slope of the LV end-systolic P/V or the Ees and increased LV volume at 1 month after MI. These changes worsened at 6months in the control group. The treated group showed an increase in Ees and decreased LV volume at 6 months, with no change in LV EDPVR. This is evidence of partial reversal of LV remodeling and an increase in LV contractility after patch treatment. P/V, pressure/volume; LV, left ventricle; Ees, end-systolic elastance; LV EDPVR, left ventricular end diastolic pressure/volume relationship.

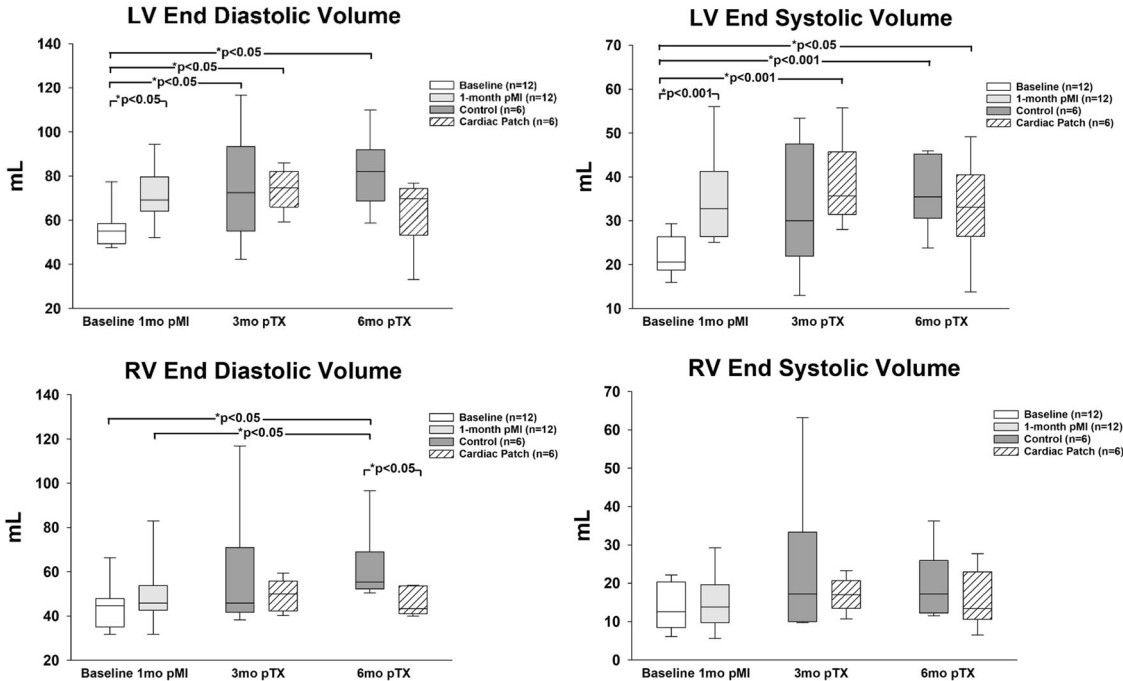

**Fig. 2 Ventricular Volumes: Left ventricular (LV) and right ventricular (RV) volumes at 3 and 6 months showed similar trends with increases in both at 3 and 6 months in the control animals, while RV end-diastolic volume trends down at both 3 and 6 months after patch placement.** LV left ventricle, RV right ventricle. Statistics performed on the data set were based on the mean ± SEM for $n = 12$ baseline animals, $n = 12$ 1-month post injury animals, $n = 6$ control animals, and $n = 6$ patch treated animals.

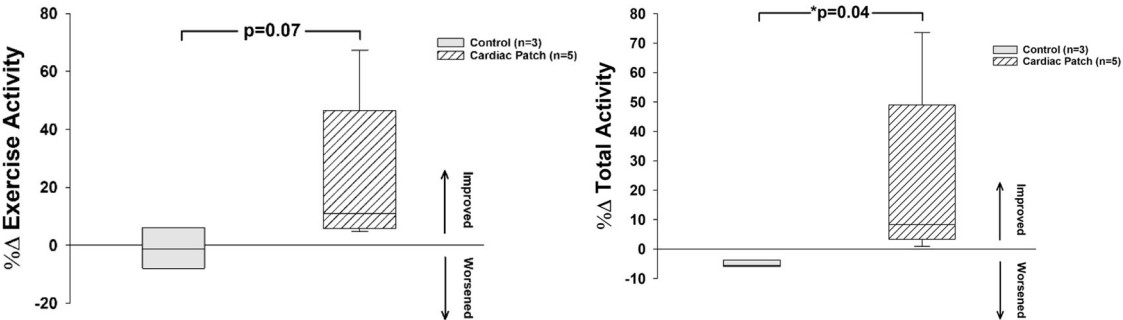

**Fig. 3 Exercise and Overall Activity: All swine in this study had activity monitored with FitBark® collars 24/7.** There was a trend for exercise activity level to increase (left) and an increase in total activity (right) in the patch treated swine. Statistics performed on the data were based on mean ± SEM for $n = 3$ control swine and $n = 6$ patch treated swine, with $p = 0.07$ for exercise activity level, and $p = 0.04$ for total activity.

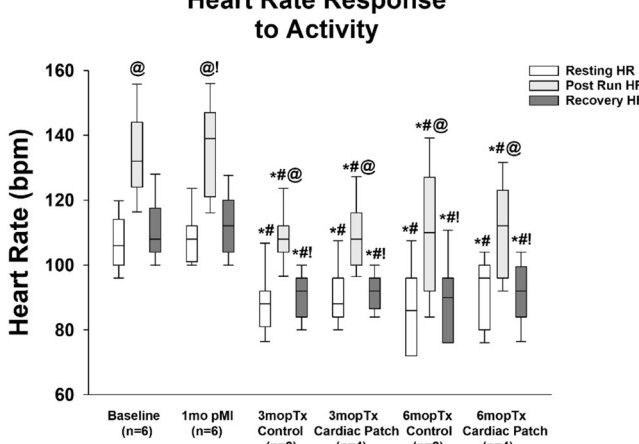

**Fig. 4 Heart rate changes with exercise.** Heart rates increased with exercise (post run) in all the swine. Heart rate decreased at 3 months after patch placement at all time points. There were no differences in heart rate at any time point between the control and cardiac patch treated swine. T-test performed and where normality failed a Mann-Whitney Rank Sum test was performed. *$P < 0.001$; # $P < 0.001$ vs 1mo pMI; @ $P < 0.001$ vs Resting Heart Rate; ! $P < 0.001$ vs Post-Run Heart Rate. These analyses were performed for $n = 6$; baseline, $n = 6$; 1 month pMI, $n = 2$; 3 month pTx Control, $n = 3$; 3 month pTx Patch, $n = 2$; 6 month pTx Control, and $n = 4$; 6 month pTx Patch.

was no mortality in control nor treatment groups during the 6-month study. Hearts treated with active patch had a reduced weight as compared to the control treated group showing that the patch reduced pathologic hypertrophy (Table 1). This reduction in hypertrophy correlates with the reduction in LV volumes, providing evidence of the benefit of the patch at a structural level. Necropsies of the thoracic and abdominal organs showed no abnormal tissue growth. Histopathology showed cardiomyocytes after patch treatment resulted in improved tissue composition with increased myocyte presence, decreased scar, and restored wall thickness (Fig. 5).

**DSP data**. We used NanoString® GeoMx® Digital Spatial Profiling (DSP) to explore changes in gene expression resulting from patch treatment in an immune competent murine MI model at the site of injury and repair. The DSP analysis isolated transcriptomic data from specific areas of interest (AOIs) and cell types of interest at week 2. For the non-treated mice, one group was taken: SYTO$^{pos}$CD45$^{neg}$ AOIs = 9 from 3 mice. For the patch treated mice two groups were taken: 1) SYTO$^{pos}$CD45$^{pos}$

AOIs = 7 from 5 mice, and 2) SYTO$^{pos}$CD45$^{neg}$ AOIs = 8 from 5 mice.

Figure 5 shows trichrome stained hearts from a normal control, a surgical control and a MI untreated mouse heart with a dilated left ventricle and a thin scarred anterior wall, and a MI patch treated heart with a thickened anterior wall. Summary data showed an increase in anterior wall thickness (0.23 ± 0.09 mm in the MI untreated mice vs 1.08 ± 0.17 mm in the MI patch mice, $P < 0.05$).

We stained cardiomyocytes (desmin), immune cells (CD45), and vasculature (α-smooth muscle actin (α-SMA)). In patch treated hearts, there was an immune response defined by the presence of CD45$^{pos}$ cells that were not present in the non-treated hearts (Fig. 6). Using DSP, we enriched for whole transcriptomic expression data from specific segments within the tissue. We segregated for two regions: 1) CD45$^{neg}$desmin$^{neg}$α-SMA$^{neg}$ cells (in both treated and non-treated animals) and 2) CD45$^{pos}$desmin$^{neg}$α-SMA$^{neg}$ cells (present only in treated animals). We selected only cell segments within the infarct zone by referring to the Masons Trichrome stained slides. To check the quality of segmentation, we analyzed the expression of desmin and CD45 within the segments, finding that we were enriching for PTPRC expression within the CD45$^{pos}$ segments and selecting for desmin$^{pos}$ cells (Fig. 7).

To assess tissue diversity and global changes after patch placement, we performed uniform manifold approximation and projection (UMAP) of the infarct regions in treated and untreated mice. This UMAP demonstrated a high spatial diversity within the infarct zone CD45$^{neg}$ desmin$^{neg}$ groups with the segments from both patch-treated and MI controls appearing dispersed and not forming unique clusters (Fig. 7). This suggests that there is not a major change within the CD45$^{neg}$ desmin$^{neg}$ cell groups within two weeks. Conversely, the CD45$^{pos}$ sections displayed clustering distinct from the CD45$^{neg}$ desmin$^{neg}$ segments. Quantification of the percent expression of CD45$^{pos}$ signal showed an increased presence in patch-treated MI versus MI alone (9.0 ± 2.4% vs 1.2 ± 0.3%, $P < 0.05$) (Fig. 6). The difference in the mean values of the two groups is greater than would be expected by chance; there is a statistically significant difference between the experimental groups ($P = 0.030$). The data concludes the patch treated animals had significantly higher levels of CD45 expressed in patch treated regions, than in the infarcted regions of non-treated animals.

Cellular deconvolution was used to identify major cell type changes within the DSP segments. While the changes were not significantly different within the CD45$^{neg}$ desmin$^{neg}$ segments, the CD45$^{pos}$ segments demonstrated an elevated macrophage and dendritic cell signature. Macrophage and dendritic cell relative abundances were elevated in the infarcted region after two weeks

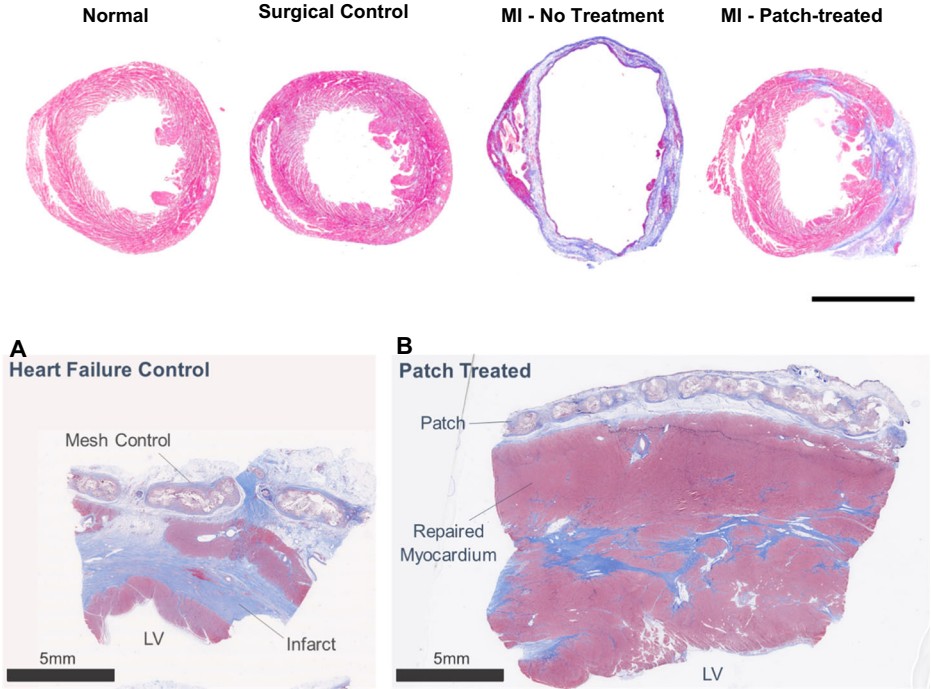

**Fig. 5 Trichrome stained cross-sections of porcine and murine hearts sections.** Masson's trichrome histopathological sections of porcine control (**A**) and patch treated (**B**) swine at 6-month study endpoint. All swine received myocardial infarction by a 90-minute balloon occlusion followed by reperfusion. Heart failure was allowed to develop over 4-weeks prior to treatment. Swine received epicardial implant of either Control (bioresorbable mesh without cells) or patch. Control tissue had a higher scar burden and a thinner LV anterior wall. Patch treatment resulted in improved tissue composition (increased myocyte presence and decreased scar) and restored wall thickness. After fixation each murine heart was transversely cut approximately 3-5 millimeters above the apex. The apex was formalin fixed, paraffin-embedded and sectioned with subsequent Masson Trichrome staining to define infarcted myocardium (blue) and healthy myocardium (red). There is essentially no difference between normal and surgical control hearts, the MI heart has a dilated LV cavity with thinned scarred anterior wall, the MI- patch shows a smaller LV cavity and thicker anterior wall compared to the MI heart. Normal = no intervention; surgical control = thoracotomies without MI or patch treatment. Abbreviation: MI-myocardial infarction. Murine heart scale bar = 3 mm.

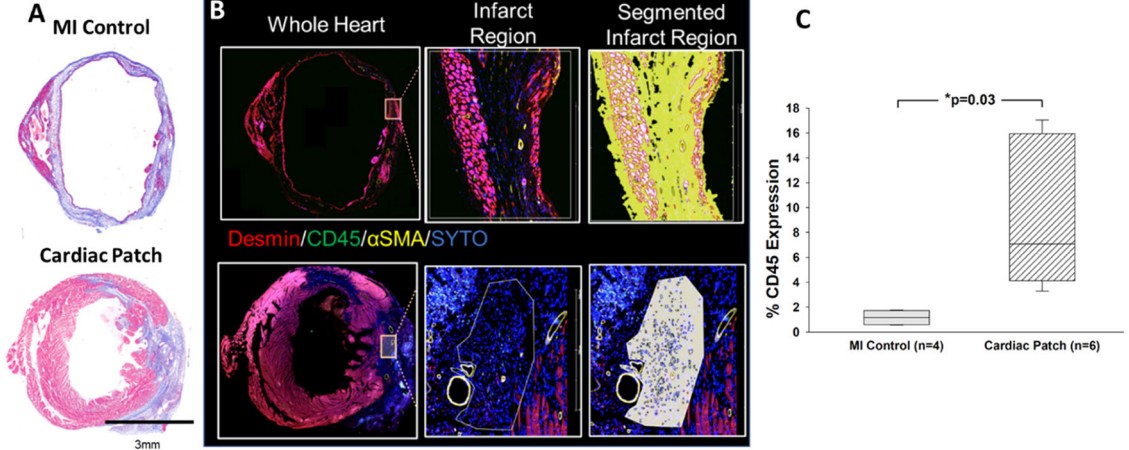

**Fig. 6 Immunohistochemistry for DSP and CD45 quantification.** Trichrome images next to the IHC stained tissue sections prepared for the DSP analysis: Desmin (red)-muscle/endothelial cells, CD 45-immune cells (green), αSMA-activated fibroblasts (yellow), SYTO-nuclear stain (blue). **A** The trichrome images show a dilated left ventricle with a thinned scarred anterior wall with decreased cardiomyocytes and minimal immune cell infiltrate. **B** The patch treated images show a smaller left ventricle with a thickened anterior wall and dramatically increased immune cell presence. The images confirm the structural changes and increased number of immune cells with patch treatment. **C** %CD45 expression per ROI is significantly elevated in Cardiac Patch treated group compared to control MI. Statistics performed on CD45 quantification were based on mean ± SEM for $n = 4$ AOIs and $n = 6$ AOIs, with $p = 0.03$.

of patch treatment compared to the untreated groups, including control and sham (Fig. 7). There were increases in macrophage relative abundance consistent with increased expression of PTPRC (CD45), CD68 and ADGRE1 (Fig. 8). Increased dendritic cell relative abundance was confirmed by examining CD11b expression. Macrophage phenotype was determined by assessing expression of known M2 macrophage markers: RETNLA and MRC1 (Fig. 8). These markers were upregulated in the infarcted region while the M1 markers NOS2 (Normalized Expression. Control: $12.73 \pm 0.99$, Patch CD45$^{neg}$: $11.12 \pm 2.57$, Patch

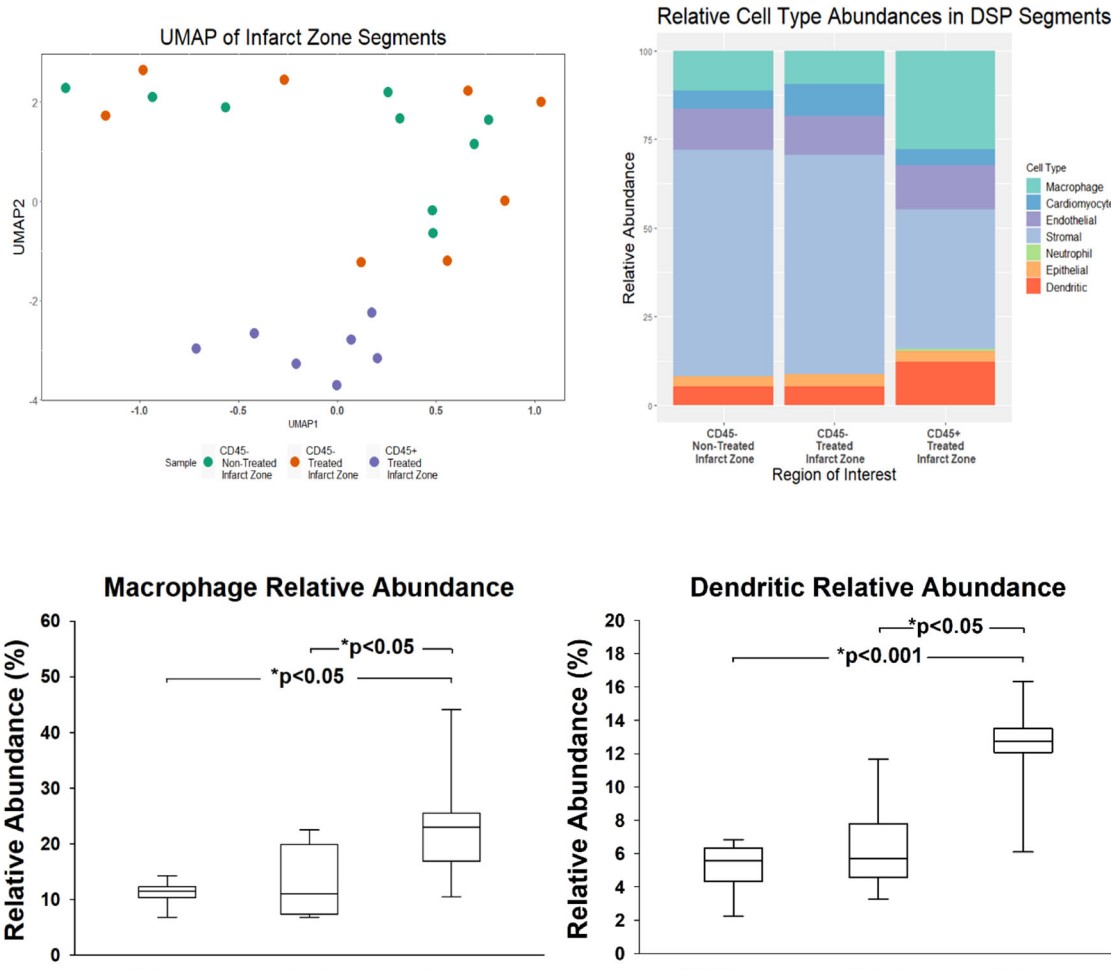

**Fig. 7 Digital spatial profiling analyses with macrophage/dendritic cell abundance.** UMAP of MI-non-treated and MI-patch-treated infarct region transcriptome profiles with distinct clustered CD45pos regions [Left]. Cell deconvolution [Right] of the CD45neg and CD45pos segments of the infarcted region showing increases in macrophages and dendritic cells in CD45pos treated infarct zone. UMAP, uniform manifold approximation and projection; MI, Myocardial Infarction. Relative Abundance (left) and Dendritic Relative Abundance (right) increase in CD45pos treated infarct zones, compared to MI control and CD45neg regions. MI, Myocardial Infarction. Statistics performed on the data were based on mean ± SEM for $n = 9$ MI Control, $n = 17$ Patch CD45neg, and $n = 16$ Patch CD45pos, with $p < 0.05$ for all indicated groups, except for dendritic cells in patch treated CD45 negative and positive regions, where $p < 0.05$.

CD45pos±: 11.04 ± 1.64, Mean ± SE) and IFN7 (Normalized Expression. Control: 21.72 ± 3.78, Patch CD45neg: 22.98 ± 5.89, Patch CD45pos±: 12.60 ± 1.83, Mean + SE) were not despite a significant increase in macrophage relative abundance. This indicates that the patch polarizes macrophages from their inflammatory M1 state to their anti-inflammatory reparative M2 state, defined by increases in RETNLA and MRC1 expression (Fig. 8).

**Conclusions from DSP analyses.** The increased presence of M2 markers in the patch zone suggests that the patch provides the chemical milieu to favor maturation into M2 macrophages increasing their number locally. It is well documented that M2 macrophages have a role in wound repair and a reparative nature. The ability of the patch to establish a immune response employing M2 macrophages may be integral to the repair response induced by the patch. This concept has been suggested as a potential mechanism of action of cell-based therapy for ischemic CHF but, to our knowledge, until now it has not been proven[17,18].

## Discussion

This study suggests that a biologic patch therapy resulted in improvements in cardiac function in an immune competent mini swine model of ischemic CHF. After 6 months of treatment, this patch improved LV pathophysiology, restored LV contractility without creating constrictive physiology, partially reversed maladaptive LV and RV remodeling, and increased exercise tolerance, without inducing ventricular arrhythmias. Myocardial oxygen consumption and the rate-pressure product did not change. There was no mortality in treated swine; all swine remained in sinus rhythm with no ventricular arrhythmias, with increased daily activity, and decreased CMR derived T2 relaxation indicating a reduction of localized edema associated with the patch, another sign that cardiac patch was well tolerated[19,20].

Treated hearts demonstrated increased CD45pos immune infiltration and increased macrophage and dendritic cell relative abundances. Macrophage gene expression of PTPRC (CD45), CD68 and ADGRE1 were up regulated in the infarcted region with treatment as compared to the untreated group. There were no appreciable CD45pos cells present in control and sham

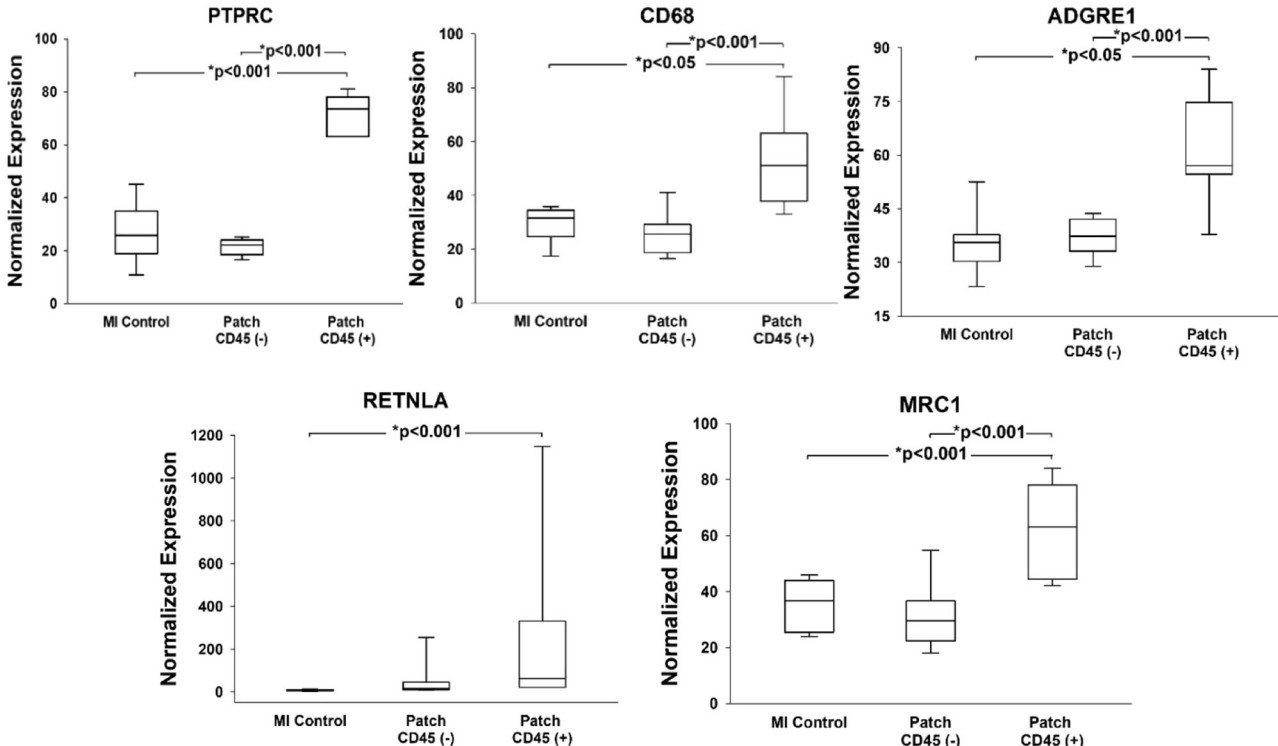

**Fig. 8 Immune cell characterization.** High macrophage abundance within CD45pos infarct region defined by increases in PTPCR, CD68, and ADGRE expression compared to lower expression in CD45neg non-treated and treated infarct zones. *P < 0.05, *P < 001. Abbreviations: PTPRC: protein tyrosine phosphatase receptor signaling molecules that regulate cell growth, differentiation, mitosis, and oncogenic transformation; CD68: Routinely used as a histochemical/cytochemical marker of inflammation associated with the involvement of monocytes/macrophages; ADGRE: Adhesion G protein coupled receptor, activated in dendritic cell development. The cardiac patch polarizes macrophages to anti-inflammatory states within the CD45pos regions as defined by increases in RETNLA and MRC1 expression, compared to CD45neg non-treated and treated infarct zones. Abbreviations: RETNLA: Alternatively activated macrophage marker for M2 phenotype; MRC1: Mannose Receptor C-Type 1 is a membrane receptor that mediates the endocytosis of glycoproteins by macrophages. Statistics performed on the data were based on mean ± SEM for n = 9 MI Control, n = 8 Patch CD45neg, and n = 7 Patch CD45pos, with p < 0.01 for all indicated groups, except for CD68 and ADGRE1 markers between MI and patch treated CD45pos regions, where p < 0.05.

animals. In addition, M2 macrophage markers RETNLA, and MRC1, were upregulated in patch treated tissue suggesting an immunomodulatory role of the patch application with polarization of M2 macrophages as a mechanism of therapeutic benefit. Interestingly, the macrophages were accompanied by dendritic cells, while far less understood in the role of cardiac repair, are associated with wound healing[21]. Based on these data, we propose that the increase in anti-inflammatory reparative M2 macrophages in the damaged heart stimulates cardiac repair and decreases fibrosis.

Our DSP analyses showed changes in gene expression of immune associated markers at the site of injury in response to treatment. This concept contrasts with a majority of the field in which immune suppression is provided to either promote integration of the therapeutic cell or extended their persistence to prolong paracrine signaling[9,22–26].

Importantly, immune suppression is a component of the current clinical trials of engineered cardiomyocytes: 1) Cell-sheet studies with allogeneic hiPSC-CMs in Japan (NCT046963282), 2) Injected allogeneic hiPSC-CMs in China (NCT03763136), 3) Injected embryonic stem cell derived cardiomyocytes at Stanford University (NCT05068674), 4) Injected iPSC-derived human engineered heart tissue in Germany (NCT04396899) and 5) Injected iPSC derived cardiac spheroids in Japan (NCT04945018). All these trials require either short or long-term immune suppression. Clinical complications of immunosuppression include, increased risk of infection, continued patient monitoring, patient compliance of taking the drugs, increased

cost, etc. Based on our data, blocking the immune system may decrease the effectiveness of cell therapy.

Our patch is a novel xenograft transplant implanted in immunocompetent rats, swine, and mice, who received no immunosuppression where we do not intend for the cells to engraft. The presence of CD 45pos in the patch treated hearts was observed. These cells are responsible for clearing the transplanted cells and do not have a role in recovery. This approach enables modulation of the immune system which as we and others have reported can promote tissue repair[27]. The use of immune suppression with cell-based therapy is intended to promote long-term persistence of the cells as an integrative approach whereas we report here the beneficial effects with a non-integrative strategy, where the transplanted cells do not integrate long-term into the host tissue[12,16]. However, previous data on immune responses to cellbased therapy show that an immune response may be necessary in repairing the heart one week after an acute MI[28–32]. These same authors showed that mononuclear cells injected into the heart 8 weeks after an MI failed to rejuvenate the scarred rodent heart[32]. These authors concluded that since they only examined mononuclear cells, "future studies are warranted using other cell-based modalities," which would include our work with hiPSC-CMs.

Arrhythmogenesis remains one of the leading causes of concern with integrative and particularly injected iPSC-CMs cell therapy[24,25]. Despite considerable work to alleviate these concerns it is uncertain when these therapies may be ready for the clinic[26]. Our data show the absence of ventricular dysrhythmias

**Table 2 Sedation, anesthesia and analgesia.**

| Species | Sedation | Anesthesia | Surgical Analgesia |
|---|---|---|---|
| Swine | Ketamine 11-33 mg/kg, IM<br>Midazolam 0.1-0.5 mg/kg, IM or Telazol 5-8 mg/kg, IM | Isoflurane 1-3% | Buprenorphine SR 0.12-0.27 mg/kg, SC<br>Carprofen 2-4 mg/kg, IM;PO;SC Bupivacaine 2 mg/kg, SC,II |
| Mouse | Isoflurane 1-3% Chamber | Isoflurane 1-3% | Buprenorphine SR Carprofen |

which is critically important for clinical translation. The application of this patch to the epicardial surface of the heart, as opposed to direct cell injections into the myocardium, maybe one of the reasons we do not see ventricular arrhythmias. It is thought that direct injection of high dosage cardiomyocyte cell therapies into the heart has the potential damage the ventricular wall and set up isolated foci of transplanted cells in the myocardial wall that maybe responsible for the generation of ventricular arrhythmias[24,33]. Further we hypothesize that our low dose, and non-integrative approach circumvents the issue of ventricular dysrhythmia formation. In our previous work, we showed the cardiac patch enhanced electrical stability in infarcted hearts with improvements in ventricular voltage and conduction velocities[12].

An important consideration to clinical work is the reliance is immunosuppression which is a component of most current clinical trials: 1) Cell-sheet studies with allogeneic hiPSC-CMs in Japan (NCT046963282), 2) Injected allogeneic hiPSC-CMs in China (NCT03763136), 3) Injected embryonic stem cell derived cardiomyocytes at Stanford University (NCT05068674), 4) Injected iPSC-derived human engineered heart tissue in Germany (NCT04396899) and 5) Injected iPSC derived cardiac spheroids in Japan (NCT04945018). All these trials require either short or long-term immune suppression. Clinical complications of immunosuppression include, increased risk of infection, organ toxicities, continued patient monitoring and blood testing with strict patient adherent regimes and increased cost. Based on our data, blocking the immune system may decrease the effectiveness of cell therapy.

With ischemia/reperfusion of the LAD, the resulting infarct often involves the intraventricular septum as well as the LV anterior wall. Epicardial placement of the patch does not provide direct contact with the septum. While this is a potential limitation, if the mechanism(s) of action involves changes in gene expression and cell signaling, those effects have the potential to reach the intraventricular septum. This concept is supported by the DSP data showing changes in gene expression in the border zone of the treated infarct and into adjacent healthy myocardium where the patch does not make direct contact with all the damaged tissue. Based on the global improvements in LV function as well as RV and LV remodeling, it is apparent the benefits of the patch are not localized to the specific local area of the heart to which it is attached.

This cardiac patch embedded with human NDFs and seeded with hiPSC-CMs improves LV contractility/function, partially reverses LV and RV remodeling, improves exercise activity, does not cause ventricular dysrhythmias nor increase myocardial energy utilization in non-immune suppressed Yucatan mini swine with ischemic CHF after 6 months of treatment. These data combined with our previous work show no safety concerns with xenograft transplants and since the patch increases blood flow to the infarcted myocardium, suggest that this patch has potential application as a treatment in patients with poor LV function due to an ischemic cardiomyopathy. The mechanisms of action are related to localized immune mediated changes in gene expression at the site of injury and repair that polarize macrophages from their proinflammatory M1 state to their anti-inflammatory reparative M2 state. While we have only tested this approach in

a preclinical model of ischemia induced decreases in LV function, the data support the concept that this patch could effectively treat any cardiac disease that is manifested by a loss of functioning cardiomyocytes.

## Methods

**Animal studies.** All animal work was performed with oversight of the University of Arizona.

Institutional Animal Care and Use Committee (IACUC) and the University of Arizona Animal Care (UAC) veterinary staff. The IACUC oversees the University of Arizona's animal care and use program and is responsible for reviewing and approving all activities utilizing vertebrate animals for research, teaching, and testing. Compliance Information: USDA, Class R Research Facility, Registration Number: 86-R-0003, Expiration Date: August 24, 2022. NIH/OLAW Assurance Number: D16-00159, Expiration Date: August 31, 2023 and AAALAC International Accredited since 1969, Accreditation Number: 000163, Status: Continued Full Accreditation.

Animals were observed at minimum once daily, with increased monitoring after surgical procedures. Light/dark cycles are 12 hr/12 hr, standard mini-swine chow was given morning and afternoon and mouse chow was provided ad lib. Fresh water was available ad lib. Cage enrichment is provided to all species.

Swine were housed separately in bedded runs with visual and physical contact with neighboring animals, Swine enrichment including frequent staff interaction for training, skin care, internal cardiac monitor (ICM) checks, collar checks, treadmill runs and play. Mice were housed up to 5 per cage. We have complied with all relevant ethical regulations for animal use.

**Anesthesia and analgesia.** Minor procedures were performed under sedation with light anesthesia; echocardiography, CMR and surgical procedures were performed under deep anesthesia. Prophylactic antibiotics were given prior to surgical procedures. Analgesia was given for 72 hours up to 7 days after procedures. Euthanasia was performed under deep anesthesia using potassium chloride 2 meq/kg. Organ removal was performed after euthanasia (see Table 2).

**Sedation, anesthesia and analgesia.** Dosages are provided. Analgesia was given after minor and major surgeries. Light sedation and anesthesia were used for non-surgical but stressful procedures, such as echocardiography.

**Swine Physiologic Study Outline.** A total of 17 immune competent Yucatan mini swine were used for the ischemic chronic heart failure (CHF) model. One animal died during the myocardial infarction (MI) procedure, study total $n = 16$: Control $n = 6$ and Patch-treated $n = 8$. One patch-treated animal was excluded from LV and RV data due to infection/unattached patch. Animals were randomly selected for matrix only (control CHF or hiPSC-CM/NDF/matrix patch treatment. All study members were blinded to treatment.

Fig. 1 gives the timeline of study procedures. Male swine 8–14 months old at arrival, underwent acclimatization to staff and

nylon collars over a 1–4-week period. When swine were comfortable with staff members and wearing collars fitted with FitBark 2 activity monitors (FitBark, Kansas City, MO, USA), treadmill (Weslo Cadence G 5.9, Icon IP, Logan, UT) training was started. Treadmill training typically took 4-6 weeks of daily training for the swine to calmly trot on the treadmill and the team to establish the maximum rate for each animal. Baseline activity and exercise data collection was started and collected 3-6 weeks before the baseline cardiac magnetic resonance (CMR) imaging.

**Cardiac Magnetic Resonance (CMR) imaging.** The CMR imaging was performed in the University of Arizona Translation Bioimaging Resource's Siemens Skrya Magnetom 3 Tesla MRI system (Skrya Magnetom, Erlangen, Germany). The images were analyzed by an independent blinded radiologist, to define changes in heart volumes and infarct size, as reported previously[21]. In brief, the swine were anesthetized as described above and continuously monitored by UAC veterinary staff. A Philips Expression IP5 (Koninklijke Philips, NV USA) system was used for gating. The T2 mapping sequences were obtained with Circle-Cardiovascular Imaging 42 (CVI42, Calgary, Canada) from each swine timepoint. Two animals were excluded from CMR data due to heart rates >30 beats/min than baseline rates.

**Hemodynamic & pressure/volume data acquisition.** Hemodynamic and pressure/volume (P/V) data were obtained with a conductance catheter (Millar Ventri-Cath 507, Millar Instruments, Houston, TX, USA) through the Mikro-Tip® pressure volume system, PowerLab16/32 with ECG Bio AMP and Lab Chart Physiologic Software (AD Instruments, Colorado Springs, CO, USA). Physiological data and P/V loops were collected at baseline prior to the creation of the MI, prior to treatment 4 weeks after MI and 6 months after treatment, prior to the endpoint study. Vital measurements, ECG, arterial pressure, and heart rate were monitored continuously. The Ventri-Cath 507 was pressure calibrated prior to insertion. Volume and saline calibrations were performed after insertion and electrode adjustment within the left ventricle. To obtain LV pressure measurements, 0.35 guide wire was advanced via a femoral artery sheath into the left ventricle and a guide catheter (IMA 90 cm) was advanced over the guide wire. The guide wire was removed and a LV angiogram with Opti-Ray 200 (Guerbet America, LLC. Richmond Heights, MO, USA) contrast agent was injected through the guide catheter. To obtain changes in LV loading, the inferior vena cava was temporarily occluded, reducing systolic blood pressure up to 30 mmHg and repeated 3-5 times, to obtain 2-3 consistent P/V loops. Blood pressure and heart rate were allowed to return to baseline after each P/V loop. Post processing of the P/V loop data provided the end-systolic elastane (Ees) and LV end-diastolic pressure relationship (LV EDPVR).

The rate pressure product was calculated (heart rate X systolic arterial pressure) and myocardial oxygen consumption ($MV0_2$) was calculated according to the regression formula[34,35].

$$MV0_2 = C0 + C1 \times HR \times LVSP + C2 \times HR \times + dP/dt$$

Where $C0 = 0.5355$

$$C1 = 0.1282 \times 10^{-3} \text{mmHg}^{-1} \text{min}$$

$$C2 = 0.2210 \times 10^{-5} \text{mmHg} - 1 * \text{s min}$$

**Quality of life activity monitoring.** Each animal was fitted with a collar that contained a FitBark™ activity tracker collars were placed on the animals 8-12 weeks prior the MI procedure for baseline data and were worn for the 6-month duration of the study. FitBark™ 2 activity collars utilize a 3-axis accelerometer to measure activity counts "Bark Points" an arbitrary unit of measure assigned by the manufacturer to quantify activity levels. The activity counts are acquired on a minute-by-minute basis until endpoint for each swine. Data were retrieved from the collars every 24-72 hours and analyzed using Microsoft Excel pivot tables.

**Treadmill activity.** Prior to initiating treadmill test, the pigs were distracted with peanut butter and resting heart rate was acquired using a stethoscope. A six-minute treadmill test was performed with the swine enticed onto the treadmill with treats (small bits of fruit or vegetables). The treadmill started at 1mph and slowly increased over 1 minute until the pre-established maximum was reached for each animal. The maximum rate was then held for 5 minutes. During the treadmill test, the swine were closely monitored by two technicians to ensure swine safety. If the swine left the treadmill and refused to return the trial was stopped for the day. The distance the swine traveled was recorded. Immediately following the treadmill test, the heart rate was taken and repeated 5 minutes later during post-test recovery.

**Arrhythmia monitoring by insertable cardiac monitors (ICM).** The ICM Reveal LINQ™ insertable cardiac monitors (Medtronic, Minneapolis, MN, USA) were implanted subcutaneously along the left scapula 4-8 weeks prior to the MI procedure to continuously monitor for arrythmias. These ICMs were left in place for the duration of the study and interrogated every 4 weeks, or weekly if the battery was low. An independent electrophysiology nurse practitioner (MKP) reviewed the data and reviewed any recorded arrythmias with an electrophysiologist.

**Porcine and murine patch production, characterization, and scalability.** A proprietary bioresorbable mesh 5cm in diameter with a thickness of 2mm was sterilized using ethylene oxide and allowed to degas for two weeks. Previously expanded neonatal dermal fibroblasts (NDFs) and hiPSC-CMs were thawed and cultured to incorporate them into the sterile matrix disks. Cultures were maintained at 37°C and 5% $CO_2$ for 30 days. We used hiPSC-CMs that undergo a 20-day differentiation process and express greater than 90% cardiac troponin T CMs. The human NDFs were obtained from human foreskin and were expanded for patch fabrication. Patches contain a cardiomyocyte-to-fibroblast ratio of 1:1 to 2:1. The seeded matrix disks (patches) were cultured in RPMI and B27 with medium being exchanged every 24-72 h. The patches were visually assessed for the fibroblasts having completely filled the interstices of the mesh and synchronous contractions observed with this cellularization. At completion of culture cardiac patches were cryopreserved. Porcine patches were placed in Origen CS50 (Austin, TX, USA) cryopreservation biobags, sealed, filled with Cryostor® CS10 (Biolife Solutions®, Bothell, WA, USA) cryopreservation solution, and subjected to controlled and stepwise cooling process to -90°C. The bags were transferred to <-170°C and stored in vapor phase liquid nitrogen. Murine sized patches were obtained from larger patches maintaining the 1:1 to 2:1 ratio as described above. Prior to implantation, the cardiac patch was thawed using a 37°C water bath and maintained in RPMI with B27 at 37°C and 5% $CO_2$. Visual observations of synchronous contraction, integrity of the biomaterial, cell composition, and morphology were used as a characterization standard in this study. Prior to transfer to the surgical suite, the medium was removed and replaced with 1X Phosphate-Buffered Saline (PBS). This approach allowed us to successfully scale the patch and show functional benefit in vivo as displayed by the data in this manuscript.

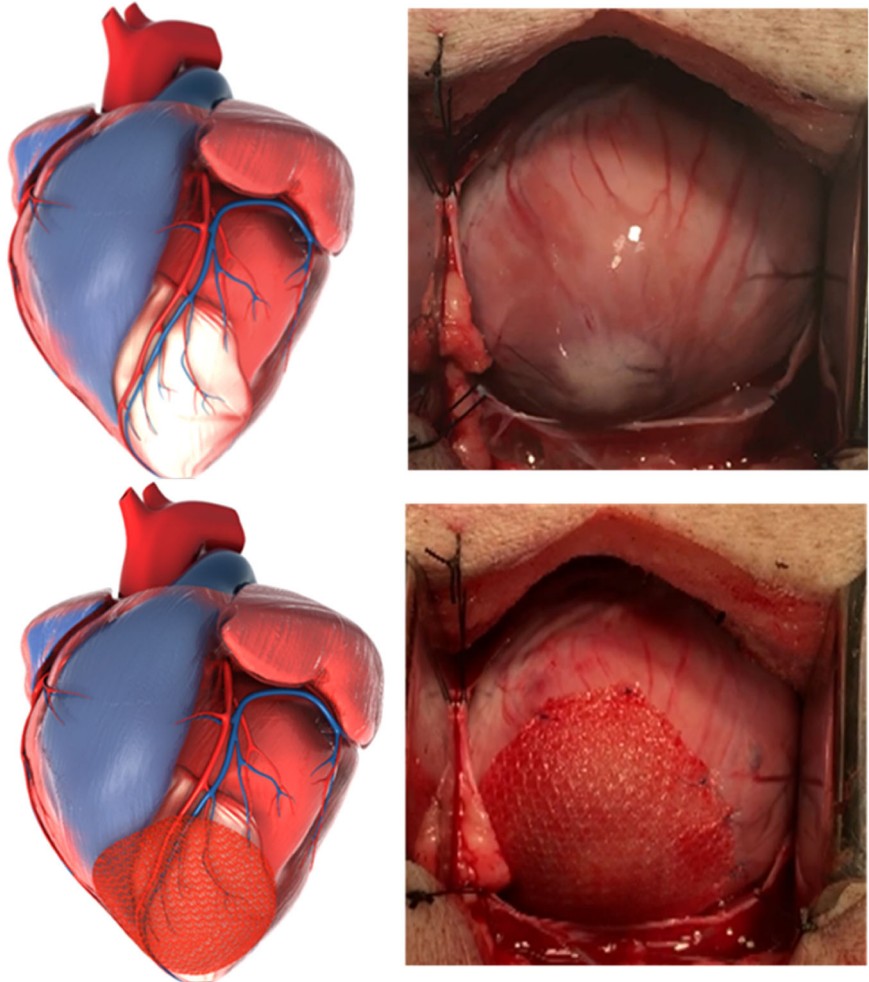

**Fig. 9 Computer models are produced from CMR images and compared to intraoperative views of patch placement.** This figure includes a computer model of an infarcted swine heart that was generated from Cardiac Magnetic Resonance (CMR) imaging. The top left image shows the infarcted region in white. In intraoperative view of the same swine, top right, only the infarcted apex is visible. The cardiac patch is placed to cover as much of the scarred area as possible. The bottom images show the computer model placement [Left] and the actual placement on the heart [Right].

The patch surface area was expanded allowing a standardized cell/cm² dose for the intended recipient (e.g., mouse, rat, swine), while the thickness remained constant and thin enough for perfusion. The increase in surface area is a common principal standard in cell culture work to generate more cellular material. Because the patches remain relatively thin (2.0mm), the patches receive adequate nutrients and perfusion by standard cell culture techniques. While the surface area is varied based on the animal model, this thickness is consistent across all studies[12,16,36].

**Implantation of cardiac patch**. A cardiac surgeon (KF) exposed the heart through a median sternotomy. The pericardium was opened and retracted with stay sutures. The cardiac patch was implanted with the cell side down onto the epicardial surface of the heart, whereas the inert control matrix was unidirectional. The patch or matrix were applied directly over the infarcted area of the left ventricle using interrupted sutures (5-0 prolene), along the border of the patch or matrix. We provided the surgeon with a CMR image that defined the extent of the infarct so he could place the patch to appropriately cover the damaged area of the left ventricle, because the extent of the infarct is not visible on the epicardium (Fig. 9). Fig. 9 shows a computer model from CMR representing the infarct and the infarct as seen in the operating room. The infarcted area extends beyond what is visible in the

operating room and the patch is placed to cover the entire region. Fig. 9 includes the view in the operating room after the patch has been sutured to the heart, covering the infarcted area of the myocardium. The pericardium was closed using Gore® Preclude® pericardial membrane (W.L. Gore & Associates, Inc., Flagstaff, AZ, USA). The sternum was closed with #1 prolene. The muscle and skin layers with absorbable sutures. The entire procedure took 20–30 minutes skin-to-skin.

**Euthanasia and necropsy**. At the 6-month study endpoint hemodynamics and P/V data were collected before swine were euthanized with potassium chloride (KCl) 2 meq/kg administered intravenously. The thoracic and abdominal cavities were inspected. The heart, lung, liver, gall bladder, stomach, pancreas, intestine/colon, spleen, kidneys, and bladder were collected, observed for any abnormalities, cleaned, and weighed.

**Histology**. The excised, cleaned heart was cooled at –80°C for 10 minutes then sectioned along the short axis from apex to base in 1-2cm sections, providing 4-5 total sections. Approximately 1 cm x 1 cm samples were taken from the apical cap, left ventricle, right ventricle and septum. The samples were fixed in 10% buffered formalin and transferred to 70% EtOH prior to undergoing tissue processing, paraffin embedding and sectioning. Slide

sections were stained with hematoxylin & eosin, Masson's tri-chrome and immunohistochemistry for human mitochondria. Tissue preparation was performed by Allele Biotechnology (San Diego, CA, USA) or StageBio (also referred to as Alizee, Mt Jackson, VA, USA) and imaging and pathological assessment was performed by StageBio. Analysis was performed on heart sections from each pig at study endpoint.

**Methods for control of bias**. Swine were randomly assigned to cardiac patch treatment or inert mesh only control. All the data collection and analyses were done by individuals blinded to the therapy. The CMR images and image interpretation were per-formed by third party cardiac radiologists blinded to treatment. Data were collected and tabulated in excel spreadsheets for eva-luation again in a blinded manner.

**Statistical Methods and Reproducibility**. Cardiac magnetic resonance imaging and hemodynamic data were evaluated using the paired students t-test between patch treated and inert mesh only control treated groups. The hemodynamic data were ana-lyzed using the students paired t-test, where normality failed, Wilcoxon Signed Rank test was performed.

**Digital Spatial Profiling (DSP)**. The NanoString GeoMx DSP analysis system in the murine model of MI was used to obtain transcriptomic data of four major cell populations of immune cells, fibroblasts, cardiomyocytes, and patch cells in a murine model of MI.

**Murine study**. Male C57BL/6 mice aged 9 weeks were infarcted via left coronary artery (LCA) ligation, as described previously[37]. In brief, using isoflurane induction, intubation, ventilation with isoflurane anesthetic, a left lateral thoracotomy was done, peri-cardium opened, and a permanent ligature was placed around the left coronary artery. The chest wall was sutured closed while the lungs were inflated, the muscle and skin layers were sutured separately. The mice were recovered for 3 weeks, anesthetized for cardiac patch placement on the epicardium. The mice were randomly assigned to MI control (no patch) ($N = 3$) and MI with patch ($N = 5$).

**Histology and Immunohistochemistry (IHC)**. The mice were anesthetized and euthanized via apical injection of KCl. The heart was excised, rinsed, weighed, perfused with 10% neutral buffered formalin (NBF) and stored over night at room temperature. The following day the heart was cross sectioned into base and apical halves and put in 70% EtOH at 4°C. The base and apical heart halves were paraffin embedded and sectioned.

Following trichrome staining and imaging of mouse hearts, ImageJ was used CD45$^{pos}$ cells anterior LV wall thickness. These measurements were taken in ImageJ by opening said brightfield image in the software, scaling the image to the appropriate units by converting pixels to millimeters (mm) using the "set scale" feature after drawing a line to measure the scale bar in the software. After setting the scale, another line was drawn in the software from the endocardial to the epicardial surface of the anterior wall of the left ventricle and measured using the "measure" tool in the software. Statistical analysis was conducted using SigmaPlot 12.5 software.

Following Immunofluorescence staining of mice hearts for CD45, ROIs from treated and untreated groups were taken for subsequent analyses and quantification of percent CD45 expres-sion from infarct and patch treated regions. This expression was calculated by exporting only the CD45 channel to ImageJ where the image was changed to 8-bit and the threshold set to capture the original fluorescent signature from the CD45 in the DSP software. Once the image was grayscale, it was converted to binary by selecting "process" and then "make binary". This binary image gives black and white surfaces that the software can delineate between. From here, percent area measurements from the individual ROIs were taken by the software by selecting "analyze" then "measure", area fraction must be checked using "analyze" and "set measurements" Area fraction will output the percent area in the imaging field (ROI) taken up by the white space in the binary image, which is indicative of the CD45 expression by the tissue. These percent area values were taken from the infarcted region of the MI treatment group and from the patch treated regions of the MI + Patch group. ROIs for the experimental group were averaged between animals to give n values. A t-test was performed ($P < 0.05$).

For the DSP analysis, tissue sections were prepared as recommended in the GeoMX DSP Manual Slide Preparation. In short, the tissues were deparaffinized and rehydrated as recommended followed by antigen retrieval and blocking steps. In this study we utilized CD45(Rat monoclonal (30-F11) Alexa Fluor® 647), desmin (Desmin (RD301) Alexa Fluor®594), SYTO, and α-SMA (ab184675 Alexa Fluor® 488) which were incubated overnight Proceeding the next day the slides were washed with Tris buffered saline and tween 20 (TBS-T) three times for 10 minutes each wash and covered with paraformaldehyde and incubated for 30 minutes. Lastly SYTO nuclei stain was completed as recommended by the manufacturer. The slides were submerged in 1X TBS-T at 4°C until GeoMX analysis.

**NanoString GeoMX analysis**. The tissues were sent to Nano-String DSP Technology Access Program at the NanoString facility (Seattle, Washington) Digital spatial profiling protocol was completed as previously outlined[38]. In brief, with the use of immunofluorescence to determine the morphology of the tissues, regions were selected for transcriptomic analysis.

Oligoconjugated RNA probes were used for detection and quantification of transcripts for the selected regions of tissues. Within the GeoMX select regions were defined with diameters with $760 \times 660$ pixels with minimum nuclei count of 50. Once the regions were selected, the oligo barcodes were cleaved with UV light. The cleaved oligos were dispensed into a 96-well plate and underwent PCR amplification. Once amplified, library construc-tion and next generation sequencing were completed to provide the raw count of transcripts. This process was completed for each region of interest (ROI). At present, digital spatial profiling with the NanoString GeoMX system is limited to human and murine antibodies.

**GeoMX analysis**. To determine areas of interest, NanoString research staff worked with our team to define areas labeled: remote healthy myocardium, border tissue, infarcted tissue, and cardiac patch in the treated groups. Each region was analyzed based on expression of cell markers. The strategy for selecting regions and determining transcriptomic profiles were gathered from desmin$^{pos}$ cells and the remaining unlabeled cells, (CD45$^{neg}$, α-SMA$^{neg}$-, desmin$^{neg}$) in the border region and untreated infarct region. In patch treated animals, the border region transcriptome was segregated by desmin$^{pos}$ cells and unlabeled cells similar to the control group. The infarcted region was segregated by a mix of CD45$^{pos}$ and CD45$^{neg}$.

**Statistics and reproducibility**. The University of Arizona Bioinformatics Core assisted with the statistical analyses. After initial raw counts, each segment and probe were subjected to quality control. The data was normalized using third quartile

normalization. A linear mixed model statistical test was performed. After normalization by Q3 (DSP manual) and filtering, the data was log10 transformed. The ROIs underwent cell deconvolution to identify immune cell type diversity. Significance for gene expression and cell abundance comparisons were calculated by one-way ANOVA.

**RStudio packages.** The packages used for quality control and normalization were GeomxTools (Version 3.2.0)[33] and NanostringNCTools (Version 1.6.0)[34]. Spatial deconvolution was achieved with SpatialDecon (Version 1.8.0)[35].

**Reporting summary.** Further information on research design is available in the Nature Portfolio Reporting Summary linked to this article.

## Data availability

The datasets used and/or analyzed during the current study are available from the corresponding author on reasonable request. All data has been made available via the provided repository[39]. Numerical Source data for all plots in the manuscript can be found at https://doi.org/10.25422/azu.data.24253825.v2.

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

## Acknowledgements

We thank Drs. Taben Hale, Marc Alan Pfeffer, Sam Kim, and David Bull for their review and suggestions. We thank Amal Anilkumar, Natasha Barton, Bailey Buchanan, Luke Ciulla, Kyle Cook, Tyler Dennis, Andrew Jauregui, Danielle Spencer, and other members of the Goldman Laboratory for their help with the collection of the data. This work was supported by a grant from the Arizona Biomedical *Research Commissi*on, the WARMER Research Foundation, the Sarver Heart Center, and the University of Arizona.

## Author contributions

L.J.J.: Instrumental in entire study plan, directed the swine work, reviewed all the data, helped write the manuscript, reviewed, and approved the manuscript. G.A.: Help plan the mouse study, directed the digital spatial profiling study, analyzed all the data, reviewed all the data, helped write the manuscript, reviewed, and approved the manuscript. F.J: Helped plan the mouse study, directed and performed the analyses of the digital spatial profiling study, reviewed all the data, helped write the manuscript, reviewed, and approved the

manuscript. R.J.: Analyzed the magnetic resonance imaging (MRI) data, prepared the MRI images for the surgeon, reviewed all the data, helped write the manuscript, reviewed, and approved the manuscript. D.S.: Directed all the animal work, helped analyze all the data, reviewed all the data, helped write the manuscript, formatted all the figures in the manuscript, formatted the manuscript, reviewed, and approved the manuscript. W.S.: Initiated the digital spatial profiling work, helped plan the mouse studies, performed the initial analyses, reviewed, and approved the manuscript. F.K.: Designed swine surgical approach, performed all the swine surgeries, reviewed and approved the manuscript. G.G.: Analyzed all the data, completed the statistical analyses, assisted in the swine surgeries, reviewed, and approved the manuscript. L.L.D.: Performed the cardiac catheterization in swine, hemodynamic studies, created the myocardial infarctions, reviewed, and approved the manuscript. A.R.: Planned the MRI studies and their analyses, read all the MRIs, reviewed, and approved the manuscript. A.T.: Performed all second reads on MRI studies, helped analyze the data, reviewed, and approved the manuscript. M.A.: Assisted in all surgical procedures and hemodynamic studies in all swine procedures including surgery and cardiac catheterizaions, reviewed, and approved the manuscript. S.J.: Performed all the myocardial infarctions in mice, performed all the patch implantations in mice, analyzed the data, reviewed, and approved the manuscript. P.M.K.: Read and interpreted all the Internal cardiac monitor recorder ECGs, reviewed, and approved the manuscript. M.T.: Advised the team on all electrophysiology issues pertaining to changes in conduction in the swine, reviewed all cardiac monitor recorder ECGs, reviewed, and approved the manuscript. B.M.: University of Arizona Information Technology Services Statistician, directed all the statistical analysis, reviewed, and approved the manuscript. D.B.: Performed the quantitative analysis of anterior wall viable myocardium, CD45$^{pos}$ cells and IHC proliferative markers after patch placement, reviewed and approved the manuscript. M.M.: Directed the planning and analyses of immune responses, reviewed and approved the manuscript. P.A.C.: Directed the gene activation studies in the digital spatial profiling analyses, reviewed, and approved the manuscript. Z.M.: Advised team on macrophage activation studies, reviewed all the hemodynamic and immune response studies related to immune responses in polarizing macrophages, reviewed, and approved the manuscript. B.A.: Advised the team on macrophage activation studies, reviewed all the macrophage gene activation and antibody studies, reviewed, and approved the manuscript. K.J.W.: Instrumental in the study plan, coordinated all components of the study, helped define the ROIs for DSP analysis, helped write the manuscript, reviewed, and approved the manuscript. G.S.: Responsible for all components of the study including obtaining funding, study design, and study management. Principal author on the manuscript, wrote the initial draft, made final decisions on all aspects of the study, reviewed, and approved the manuscript.

## Competing interests

Drs. Goldman, Koevary, Lancaster and Ms. Sherry Daugherty have disclosed a financial interest in Avery Therapeutics, Inc. to the University of Arizona. In addition, the University of Arizona has a financial interest in Avery Therapeutics, Inc. These interests have been reviewed and are being managed by the University of Arizona in accordance with its policies on outside interests. The work outlined in this report were the basis of forming a commercial entity, Avery Therapeutics. Drs. Goldman, Koevary, Lancaster and Ms. Sherry Daugherty have disclosed a financial interest in Avery Therapeutics, Inc. to the University of Arizona. In addition, the University of Arizona has a financial interest in Avery Therapeutics, Inc. These interests have been reviewed and are being managed by the University of Arizona in accordance with its policies on outside interests. All other authors declare no competing interests.
