## [Peer Review File · Communications Biology]

Reviewers' comments:

Reviewer #1 (Remarks to the Author):

This article reported the application of an allogeneic cardiac patch seeded with iPSC-CMs and fibroblasts to treat chronic heart failure in a swine model. The authors reported the improvement of LV contractility, partial reversal of ventricular remodeling, and increased exercise tolerance. Mechanistically, the authors suggested that the regeneration of new cardiomyocytes and immune modulation may play a role in the observed benefits. Overall, the reviewer finds this translational study interesting. But certain claims may need to be strengthened with additional experiments and analysis. Details are as follows:

Main:

1. In figure 4, the authors claimed that there was a regeneration of cardiomyocytes based on the histology data. While this could be an interesting and encouraging observation, more data are required to support this claim. Is this observation consistent across different animals treated with the patch? Or only some animals? Can the authors perform some statistical analysis of the images? It would be a good idea to perform more in-depth molecular characterization to understand the source of the new CMs (e.g., staining for proliferation markers)
2. It seems the pigs at the 6-month time point developed cardiac hypertrophy and dysfunction reflected by changes in parameters such as increased heart rate and increased systolic and diastolic volume. Interestingly, LVEF seems normal. Is this still an appropriate model for CHF with reduced EF or it is more like HFpEF at this stage? In particular, the control animals seem to have gained a lot of weight and obesity is conducive to HFpEF. maybe it will take a longer time for the EF to drop.
3. Starting from figure 5, the authors transitioned to the mouse model. While the efforts are appreciated, it's difficult to interpret the results of the paper as a whole. Specifically, can we use the results of molecular analysis from the mouse to explain the benefits of the treatment in the swine model? What is the rationale for not performing the DSP/IF staining analyses on the swine hearts?
4. Characterization of the patch prior to/post-transplantation seems insufficient. For instance, how many iPSC-CMs and NDFs are on the patch at the time of transplantation? What is the purity/maturity/age of the iPSC-CMs? How long did the cells survive after transplantation?
5. Since immune modulation is proposed to be a potential mechanism for the therapeutic benefits of the patch, would it be important to have a control group where the animals only receive the immune suppressants without the patch? This would help us dissect the confounding factors for mechanistic understanding.

Minor:

1. Figure 1 seems to have been distorted.
2. For figure 2, can you show data from more than 1 animal?

Reviewer #2 (Remarks to the Author):

The authors investigate the efficacy of patch implantation of hiPSC-derived cardiomyocytes with neonatal dermal fibroblasts to treat ischemic heart failure. They test the efficacy of the patch

treatment in an immune-competent swine model in improving cardiovascular function. Furthermore, they evaluate the mechanisms followed by the patch treatment in a mouse model. The in vivo results show promise however, the major re-writing and some clarifications on the reported data is required.

- 1) The abstract mentions a rat model, a swine model and mouse model. However, the results only discuss the swine and the mouse model, but not the rat model. Is that referring to a previous study? That should be mentioned in the abstract and referenced in introduction.
- 2) Introduction paragraph 2: Reference 3-4 – The authors should define the limitations. What is the current approach of cell therapy lacking – how is the regenerative patch investigated by the author hypothesized to overcome those limitations?
- 3) The introduction overall needs to be restructured- 1st paragraph introduces the disease effectively, followed by current treatment in clinical trials- however those limitations are not discussed effectively. The authors directly talk about the research presented in the paper without motivating it enough – instead the last line of the introduction justifies the use of NDF in the patch. The 3rd paragraph should come last in the introduction listing what are novel techniques and findings presented in the paper.
- 4) Introduction 4th paragraph introduces the term DSP without expanding what it is. “The DSP analysis of hearts two weeks”
- 5) The language throughout the paper could be more formal – for example in Introduction 4th paragraph “the though is that”. Furthermore, the authors state ‘our hypothesis is the cells on the patch are activating mechanisms that are not present in the adult heart’ – hypothesis is stated where it is tested against in the paper- however here it is more of a speculative statement.
- 6) The results are written like methods- where a method is stated and then the results from it are just stated. They are not written in a story where the goal of each experiment is clearly stated, in addition to the design and significance of the method and results.
- 7) Terms like ‘Significantly changed’ – what is it? Increase or decrease? P-value? What is the difference? significance of difference?
- 8) MRI: why is the 3-month data not shown? The figure is not referenced. The results are stated very casually- “there were increases” – needs to be specific and pointing towards the exact graphs.
- 9) Why is the 3-month data discussed multiple times if not shown?
- 10) ‘Data not shown’ is mentioned for a lot of things – should be included in supplementary if being discussed in the paper – otherwise should not be mentioned.
- 11) Informal language-‘Arrhythmia Monitoring and Activity levels’ results section: “The fact heart rate was not different in the patch treated group with exercise is confirmatory evidence that the patch did not create constrictive physiology” It’s not a fact, it’s an observation which needs to be shown also.
- 12) Figure 3: “Did not do statistical analysis” –this should not be acceptable- report if it’s not significant. Or do not mention at all if not performing statistical analysis.
- 13) Figure 4 – Value of the scale bar not mentioned
- 14) Figure 5 and 6 do not have scale bar. Figure 6 needs to be labelled with A,B,C and so on. The authors claim that MI-patch-treated hearts had higher CD45+ cells – however the images presented do not show that – the authors should show the channels separately to represent the change in CD45+ cells. Furthermore, the immunostains should be quantified and statistical analysis should be performed.
- 15) For the whole DSP analysis, the number of mice for each condition is not stated anywhere.
- 16) The discussion can also be organized into a few paragraphs to convey the key messages succinctly. Currently, it is divided, and too long. It is hard to follow.

Reviewer #3 (Remarks to the Author):

This paper reported the beneficial effects of using a cardiac patch, which was comprised of hiPSC-CM and hNDF, to treat immune competent mini swine ischemic CHF model, and further explained that this therapeutic effect was achieved through immunomodulation. It is of interest to peers in this field to have a better understanding of the mechanisms of cell therapy in the treatment of CHF. The well-conducted PV-Loop study is a very strong evidence demonstrating the beneficial effects of these cell-

based heart repair. However, the title of this paper is misdirecting, because researchers were not achieving their goal through immunomodulation, and no supporting data was provided to testify whether directly manipulating the immune microenvironment affects therapeutic outcomes.

The figures provided in this manuscript were not carefully prepared. For example:

1. Figure 1 is unreadable.
2. In Figure 2, the title of the x-axis is "Volume (mL)_", "_ " is unnecessary.
3. In Figure 4, "5 mm" written in the scale bar can barely be seen.
4. In Figure 5, the scale bar for "3 mm" is not provided, and the image quality should be improved.
5. In Figure 6. on the left, the image looks blurred.
6. In Figure 8, the picture is squeezed.

Please revise all the figures with proper individual picture sizes and font.

Response to Reviewers Biologic Communications

Reviewer #1 (Remarks to the Author):

This article reported the application of an allogeneic cardiac patch seeded with iPSC-CMs and fibroblasts to treat chronic heart failure in a swine model. The authors reported the improvement of LV contractility, partial reversal of ventricular remodeling, and increased exercise tolerance. Mechanistically, the authors suggested that the regeneration of new cardiomyocytes and immune modulation may play a role in the observed benefits. Overall, the reviewer finds this translational study interesting. But certain claims may need to be strengthened with additional experiments and analysis. Details are as follows:

We thank the reviewer for their positive comments.

Main:

1. In figure 4, the authors claimed that there was a regeneration of cardiomyocytes based on the histology data. While this could be an interesting and encouraging observation, more data are required to support this claim. Is this observation consistent across different animals treated with the patch? Or only some animals? Can the authors perform some statistical analysis of the images? It would be a good idea to perform more in-depth molecular characterization to understand the source of the new CMs (e.g., staining for proliferation markers).

We thank the reviewer for this comment, in the revised manuscript this is now the new Figure 6. Considering the reviewer's comments, we agree that more data are required to support the claim of regeneration. To balance the claim(s) within the manuscript to the data presented we have removed the use of cardiac regeneration throughout the manuscript. Our intention for this manuscript is to report on the development and advancement of the cardiac patch into a large animal model of chronic heart failure including physiologic data paired with a semi-quantitative analysis of mechanism of action which we believe we achieved. To address the question "*Is this observation consistent across different animals treated with the patch?*" Yes, this observation of pathophysiologic improvement in the infarcted heart is consistent across mice, rats and swine (Refs #12 and 13, Figs 6, 7 and 9).

With the data presented in this manuscript we believe we have presented the therapeutic potential in a well-accepted and translatable pre-clinical large animal model of ischemic heart failure. Future work will be required to characterize the potential cellular mechanism(s) of the patch. While molecular characterization of cell-cycle re-entry, division etc. will be explored, so will the temporal aspect. This will require an extensive analysis that we propose would be beyond the scope of the present manuscript.

To respond to the reviewer's request for "*some statistical analysis of the images*," We have performed a quantitative analysis of the infarcted mouse hearts and shown increased wall thickness in the patch treated mouse hearts. "Summary data showed n increase in anterior wall thickness (0.23 ± 0.09 mm in the MI untreated mice vs 1.08 ± 0.17 mm in the MI patch mice, $P < 0.05$." (Fig 7, Page 12; Lines 1-5).

Responding to the reviewer's request for documentation of regeneration, as noted above, we have removed claiming regeneration from the manuscript. We do, however, believe the reviewers line of questioning is important and intriguing. As such we hypothesized a potential proliferation would be more likely to occur acutely after patch implant. We piloted this idea by staining the 2-week post patch treated mouse hearts for Ki67 and Aurora kinase B and found no evidence of proliferative markers in the repaired myocardium. It is still unclear to us but we hypothesize proliferation is not occurring or other methods of interrogation are needed.

2. It seems the pigs at the 6-month time point developed cardiac hypertrophy and dysfunction reflected by changes in parameters such as increased heart rate and increased systolic and diastolic volume. Interestingly, LVEF seems normal. Is this still an appropriate model for CHF with reduced EF or it is more like HFpEF at this stage? In particular, the control animals seem to have gained a lot of weight and obesity is conducive to HFpEF. maybe it will take a longer time for the EF to drop.

The increases in LV systolic and diastolic volumes are fundamental findings that occur in patients and in this swine model of ischemic heart failure after a myocardial infarction. This leads to Heart Failure with reduced Ejection Fraction (HFrEF). The EF's measured in these animals are not normal and could be considered HF mildly reduced EF (HFmrEF), a well described clinical entity. Over time, when these kinds of hemodynamically important findings are left untreated in patients, they routinely continue to dilate their left ventricles and end up with end-stage heart failure.

Enclosed are references for this swine model of HFrEF with dilated left ventricles: 1) McCall FC et al., *Nat Protoc.* 2012 Jul 12;7(8):1479-96. doi: 10.1038/nprot.2012.075; 2) Sharp et al.. *Circ Res.* 2017;121:1263-1278. DOI: 10.1161/CIRCRESAHA.117.311174; 3) Morris et al., *Sci Rep.* 2022 Mar 8;12(1):3698. doi: 10.1038/s41598-022-07611-8.

We observe decreases in EF in this swine model at one month after myocardial infarction (Table 1). The animals were not obese, the adult Yucatan mini swine normal weight range is 70-90 kg. The pigs grow during the 6 months duration of the study. We closely monitored their diets, so weight gain is consistent across groups and therefore do not predict that obesity or metabolic disease was a primary contributor to heart function in this model.

While EF is a simple and useful method to screen for patients with poor LV function, changes in LV volumes are the most accurate predictions of morbidity and mortality in patients with heart failure with references listed below.

Konstam et al. Left Ventricular Remodeling in Heart Failure Current Concepts in Clinical Significance and Assessment. *JACC: CARDIOVASCULAR IMAGING* VOL. 4, NO. DOI:10.1016/j.jcmg.2010.10.008

Mann DL. Use of ejection fraction in heart failure: a tarnished gold standard? *Journal of the American College of Cardiology - ACCEL Audio Journal* 2019. Available from: <https://www.acc.org/Education-and-Meetings/Products-and-Resources/ACCEL-Audio> [Last accessed on 20 Apr 2022].

Goldman S, Traverse JH, Zile MR, Juneman E, Greenberg B, Kelly R, Daugherty S, Koevary JW, Lancaster JJ. Perspective on the Development of a Bioengineered Patch to Treat Heart Failure: Rationale and Proposed Design of Phase I Clinical Trial. *Vessel Plus* 10 Oct. 2022;6:54. doi10.20517/2574-1209.2021.149.

3. Starting from figure 5, the authors transitioned to the mouse model. While the efforts are appreciated, it's difficult to interpret the results of the paper as a whole. Specifically, can we use the results of molecular analysis from the mouse to explain the benefits of the treatment in the swine model? What is the rationale for not performing the DSP/IF staining analyses on the swine hearts?

The reviewers' comments now refer to new Figure 7 in the updated text. We agree with the reviewer that DSP analyses would be ideal in the swine hearts. To-date, digital spatial profiling technology is only available in human and murine tissue. This is why we performed experiments in a murine model to help elucidate the potential molecular pathways triggered from placing our patch in an immune competent myocardial infarct murine model. As noted in the text, it is important to point out that we see beneficial results in this report in swine and mice and in our previous work in Sprague-Dawley rats (Ref # 12 and 13).

4. Characterization of the patch prior to/post-transplantation seems insufficient. For instance, how many iPSC-CMs and NDFs are on the patch at the time of transplantation? What is the purity/maturity/age of the iPSC-CMs? How long did the cells survive after transplantation?

We used hiPSC-CMs that express more than 90% cardiac troponin T CMs. The human neonatal dermal fibroblasts were obtained from human foreskin and were expanded for patch fabrication. The cardiomyocytes undergo a 20-day differentiation process, patches contain a cardiomyocyte-to-fibroblast ratio of 1:1 to 2:1. Our previously reported data show the cells are eliminated within 30 days after transplantation (Ref #12). This information has been added to the text (Page 24; Lines 21-22)

5. Since immune modulation is proposed to be a potential mechanism for the therapeutic benefits of the patch, would it be important to have a control group where

the animals only receive the immune suppressants without the patch? This would help us dissect the confounding factors for mechanistic understanding.

We thank the reviewer for this very important comment. In this study we wanted to observe, describe and (semi) quantify the effect of the patch treatment in comparison to no treatment. We gave no immune suppression because we needed to have a ‘healthy’ host immune system initiating an inflammatory and reparative response. At this time, our goal was to establish (as we did) if and how our treatment can change the innate mechanism. For this, a functional immune system was critical. However, one of our next goals is to understand the cellular mechanisms that mediate this response and which cells are involved in this event (both host cells and patch cells). In this setting we are planning on using specific, targeted immunosuppression and/or crosstalk blockade to try to parse-out which immune cells are involved in this mechanism and if/how crosstalk between host immune cells and patch cells is essential to induce the repair. Broad-spectrum immune suppression has not been shown to be beneficial in heart failure patients or patients following a myocardial infarction. A more targeted approach for manipulating immune response in heart failure is suggested by our study and that of others and will be the focus of future experiments to address this issue.

Minor:

1. Figure 1 seems to have been distorted.

We apologize the problems with Figure 1, we have corrected this in our new Figure 1.

2. For figure 2, can you show data from more than 1 animal?

We thank the reviewer for this comment and understand the rationale. It is difficult to display multiple animals in multiple pressure/volume loops in a single figure. We propose that the cross-sectional analysis of all swine for EDV, ESV, Ees and EDPVR in Table 1 demonstrate the physiologic benefit across multiple animals. The representative PV loops displayed in Figure 2 is an example of the total benefit, taking into account EDV, ESV, Ees and EDPVR together in relation to each other to show the physiologic extent the patch provides to the previously infarcted myocardium.

Our approach of reporting representative PV loops in a figure with the summary data presented in a table is consistent with other work with this large animal model. References below from other investigators are cited below showing this way of reporting, i.e., showing conductance catheter studies with a representative PV loop and a summary table.

MacQueen et al., A tissue-engineered scale model of the heart ventricle. *Nat Biomed Eng.* 2018 Dec;2(12):930-941. doi: 10.1038/s41551-018-0271-5. Epub 2018 Jul 23. Erratum in: *Nat Biomed Eng.* 2022 Nov;6(11):1318. PMID: 31015723; PMCID: PMC. 6774355.

McCall FC et al., Myocardial infarction and intramyocardial injection models in swine. *Nat Protoc.* 2012 Jul 12;7(8):1479-96. doi: 10.1038/nprot.2012.075. PMID: 22790084; PMCID: PMC3936356.

Sharp et al., Cortical Bone Stem Cell Therapy Preserves Cardiac Structure and Function After Myocardial Infarction. *Circ Res.* 2017;121:1263-1278. DOI: 10.1161/CIRCRESAHA.117.311174.

15) For the whole DSP analysis, the number of mice for each condition is not stated anywhere.

We apologize for these missing data, we have added the following to the text.

The DSP analysis isolated transcriptomic data from specific areas of interest (AOIs) regions and cell types of interest at week 2: normal CD45^{neg} AOIs = 9 from 3 mice, patch treated CD45^{pos} AOIs = 7 from 5 mice, and CD45^{neg} AOIs= 9 from 5 mice. (Page 11; Line 11-16).

Reviewer #2 (Remarks to the Author):

The authors investigate the efficacy of patch implantation of hiPSC-derived cardiomyocytes with neonatal dermal fibroblasts to treat ischemic heart failure. They test the efficacy of the patch treatment in an immune-competent swine model in improving cardiovascular function. Furthermore, they evaluate the mechanisms followed by the patch treatment in a mouse model. The in vivo results show promise however, the major re-writing and some clarifications on the reported data is required.

We thank the reviewer for their positive comments. As recommended, we have done a major re-writing of the manuscript and clarified the data.

1) The abstract mentions a rat model, a swine model and mouse model. However, the results only discuss the swine and the mouse model, but not the rat model. Is that referring to a previous study? That should be mentioned in the abstract and referenced in introduction.

We thank the reviewer for pointing this out. Our previous rat study was referenced in the abstract (Page 1; Line 32). As suggested, we have also added the reference our previous rat study in the Introduction (Refs #12), (Page 3; Lines 28-31).

2) Introduction paragraph 2: Reference 3-4 – The authors should define the limitations. What is the current approach of cell therapy lacking – how is the regenerative patch investigated by the author hypothesized to overcome those limitations?

We have added text to the introduction to state the limitations of References 3 and 4, as well as why current cell therapy approaches are lacking and how this product is overcome these limitations. The following has been added to the text: “These limitations included that c-kit+ cells minimally contributed cardiomyocytes to the heart and that some previous publications contained experimental flaws.” (Page 3; Lines 20-22).

3) The introduction overall needs to be restructured- 1st paragraph introduces the disease effectively, followed by current treatment in clinical trials- however those limitations are not discussed effectively. The authors directly talk about the research presented in the paper without motivating it enough – instead the last line of the introduction justifies the use of NDF in the patch. The 3rd paragraph should come last in the introduction listing what are novel techniques and findings presented in the paper.

As suggested, we have restructured the Introduction with the input from the reviewer. We thank the reviewer for his/her input.

4) Introduction 4th paragraph introduces the term DSP without expanding what it is. “The DSP analysis of hearts two weeks”

We thank the reviewer for pointing this out. As suggested, we have modified the text to include the following: “Digital Spatial Profiling (DSP), takes two concepts of Immunohistochemistry (IHC) and Next Generation Sequencing (NGS) and merges them to provide RNA or protein transcriptomics within a certain region of Interest on a tissue sample of interest. Using morphology antibodies to provide highlighted cell types, the GeoMx® instrument uses a combination of UV light and mirrors to extract these cells via oligo tags. Once these tags are lifted, they are sipped up by the GeoMx and transferred to an Illumina plate to run NGS to associate the acquired transcriptomic data with the ROI on the sample, thus, providing biologically spatially relevant data to the area. In the studies presented here, the infarcted immune-competent mouse was allowed to establish MI-induced structural changes for 2 weeks prior to patch placement.

We have added the following to the text, “Digital spatial profiling utilizes immunostaining, imaging, transcriptomics, and sequencing to allow for selection of specific areas of tissue and provides information about what the tissue looks like, what is expressed, and what the RNA transcripts are within different areas of heart, i.e., infarct zone, border zone, and patch zone in the same animal.” (Page 4; Lines 14-18).

5) The language throughout the paper could be more formal – for example in Introduction 4th paragraph “the thought is that”. Furthermore, the authors state ‘our hypothesis is the cells on the patch are activating mechanisms that are not present in the adult heart’ – hypothesis is stated where it is tested against in the paper- however here it is more of a speculative statement.

As suggested, we have rewritten major portions of the manuscript; we have changed the text from hypothesis to speculation and remove the statement: “the thought is that...”.

6) *The results are written like methods- where a method is stated and then the results from it are just stated. They are not written in a story where the goal of each experiment is clearly stated, in addition to the design and significance of the method and results.*

As suggested, we have modified Results section to define the goal, the design, and the results.

7) *Terms like ‘Significantly changed’ – what is it? Increase or decrease? P-value? What is the difference? significance of difference?*

As suggested, we have removed the term “significantly changed” and replaced it with increased or decreased.

8) *MRI: why is the 3-month data not shown? The figure is not referenced. The results are stated very casually- “there were increases” – needs to be specific and pointing towards the exact graphs.*

As suggested, we have added a new figure showing the 3-month MRI data (New Fig 3)

9) *Why is the 3-month data discussed multiple times if not shown*

As noted above the 3-month data are presented in the new Figure 3 and discussed (Page 8; Lines 5-11).

10) *‘Data not shown’ is mentioned for a lot of things – should be included in supplementary if being discussed in the paper – otherwise should not be mentioned.*

As suggested, we have removed all mention of “Data not shown” and included all the aforementioned data.

11) *Informal language-‘Arrhythmia Monitoring and Activity levels’ results section: “The fact heart rate was not different in the patch treated group with exercise is confirmatory evidence that the patch did not create constrictive physiology” It’s not a fact, it’s an observation which needs to be shown also.*

We thank the reviewer for this comment and as suggested we have changed the text to observation and added a figure showing no changes in heart rate with exercise (Fig. 5).

12) *Figure 3: “Did not do statistical analysis” –this should not be acceptable- report if it’s not significant. Or do not mention at all if not performing statistical analysis.*

As suggested, we have removed the statement: “Did not do statistical analysis” and added the statistical analyses on the figure. This is now Figure 4 in the resubmitted text.

13) *Figure 4 – Value of the scale bar not mentioned*

The 5 mm scale bar is now visible on the new Figure 6 in the revised manuscript.

14) Figure 5 and 6 do not have scale bar. Figure 6 needs to be labelled with A,B,C and so on. The authors claim that MI-patch-treated hearts had higher CD45+ cells – however the images presented do not show that – the authors should show the channels separately to represent the change in CD45+ cells. Furthermore, the immunostains should be quantified and statistical analysis should be performed.

We have made sure that the scale bars now show up on the new Figures 6 and 7, thank you for pointing this out.

To quantify the changes in CD45^{pos} signal, we used the AOIs for MI group compared to the MI+Patch group and found an increase in CD45^{pos} expression within the patch AOIs: (9.0±2.4% vs 1.2±0.3%, P<0.05). This has been added to the text (Page 13, Lines: 11-12).

15) For the whole DSP analysis, the number of mice for each condition is not stated anywhere.

As suggested, we have provided the data on the number of mice for each condition (Page; 11; Lines 13-16).

16) The discussion can also be organized into a few paragraphs to convey the key messages succinctly. Currently, it is divided, and too long. It is hard to follow.

We thank the reviewer for this suggestion. As suggested, the discussion has been reorganized and shortened to convey the key messages succinctly.

Reviewer #3 (Remarks to the Author):

This paper reported the beneficial effects of using a cardiac patch, which was comprised of hiPSC-CM and hNDF, to treat immune competent mini swine ischemic CHF model, and further explained that this therapeutic effect was achieved through immunomodulation. It is of interest to peers in this field to have a better understanding of the mechanisms of cell therapy in the treatment of CHF. The well-conducted PV-Loop study is a very strong evidence demonstrating the beneficial effects of these cell-based heart repair. However, the title of this paper is misdirecting, because researchers were not achieving their goal through immunomodulation, and no supporting data was provided to testify whether directly manipulating the immune microenvironment affects therapeutic outcomes.

We thank the reviewer for their comments, as suggested we have changed the title:

Biologically Derived Epicardial Patch Induces Macrophage Mediated Pathophysiologic Repair in Chronically Infarcted Swine Hearts (new title)

The figures provided in this manuscript were not carefully prepared. For example:

- 1. Figure 1 is unreadable.*
- 2. In Figure 2, the title of the x-axis is "Volume (mL)_", "_ " is unnecessary.*
- 3. In Figure 4, "5 mm" written in the scale bar can barely be seen.*
- 4. In Figure 5, the scale bar for "3 mm" is not provided, and the image quality should be improved.*
- 5. In Figure 6. on the left, the image looks blurred.*
- 6. In Figure 8, the picture is squeezed.*

Please revise all the figures with proper individual picture sizes and font.

We apologize for the quality of the figures. This was our error. We understood we could submit the manuscript as a word document. We think that the figures got distorted when the document was converted to a PDF by the editorial office. We have rectified this by submitting the revised manuscript as a PDF.

Reviewers' comments:

Reviewer #1 (Remarks to the Author):

Overall, the concerns raised by the reviewer are not fully addressed. In addition, the authors did not highlight changes in this revision, making it difficult to gauge the improvement in the quality.

Main:

1. The authors revised the main claim and removed all statements regarding "regeneration". However, regarding the important question "Is this observation consistent across different animals treated with the patch?". The authors referred to figures 6, 7, and 9. Unfortunately, these figures are only descriptive/representative images and do not address this concern. The only new data is described as "Quantification of the percent expression of CD45pos signal showed an increased presence in patch-treated MI versus MI alone ($9.0 \pm 2.4\%$ vs $1.2 \pm 0.3\%$, $P < 0.05$) (Fig. 9)." However, there is no such data shown in Fig.9.
2. Concern is addressed.
3. Concern is addressed.
4. Previous studies were performed on rats. The reviewer supposes the cardiac patch used for pigs would be much larger. Yet, scaling up is not always technically easy. Large tissue patches may suffer from poor perfusion, resulting in a lack of oxygen and nutrients. Therefore, it would still be necessary to characterize the patch even though it is similar to the patches used in prior rat studies.
5. Concern is addressed.

Reviewer #2 (Remarks to the Author):

No additional comments. They have made changes according to the reviews.

Reviewer #3 (Remarks to the Author):

The authors have addressed all my initial concerns, and the manuscript is much improved.

Response to Reviewers

Manuscript COMMSBIO-23-0441A: "Biologically Derived Epicardial Patch Induces Macrophage Mediated Pathophysiologic Repair in Chronically Infarcted Swine Hearts."

We thank the editor and the reviewers for their thoughtful comments.

Editor's comments: *In particular, reviewer 1 recommends providing characterization of the patch, which may be different in design and cell viability compared to smaller patches designed for rodents. Additionally, please provide the data/figure to support the quantification of CD45+ cells.*

We have addressed these critiques in our response to Rev. # 1.

Reviewers' comments:

Reviewer #1 (Remarks to the Author): Overall, the concerns raised by the reviewer are not fully addressed. In addition, the authors did not highlight changes in this revision, making it difficult to gauge the improvement in the quality.

As suggested, we have submitted a version of the manuscript with track changes to highlight the changes in this revision.

Main:

1. The authors revised the main claim and removed all statements regarding "regeneration". However, regarding the important question "Is this observation consistent across different animals treated with the patch?". The authors referred to figures 6, 7, and 9. Unfortunately, these figures are only descriptive/representative images and do not address this concern. The only new data is described as "Quantification of the percent expression of CD45pos signal showed an increased presence in patch-treated MI versus MI alone ($9.0 \pm 2.4\%$ vs $1.2 \pm 0.3\%$, $P < 0.05$) (Fig. 9)." However, there is no such data shown in Fig.9.

We thank the reviewer for their thoughtful critique. To answer the first question, yes, the observation of changes in structure and function were consistent across all animals treated with the patch as illustrated by the summary data presented in the manuscript (Table 1, Figures 3, 4, and 5). In addition, the structural changes in this report in swine and mice are the same changes documented in our previous publications in Sprague-Dawley rats (Ref # 12 and #13, Supplemental Fig. 3). This information has been added to the text (Page 6; Lines 11-13).

As requested, we added a more complete description of the method, we used to quantify the CD45^{pos} response to the text (Page 13, Lines 12-16 in the results and Page 29; Lines 11-24 in the methods). We also included a new graph in Figure 9 that displays the quantification of %CD45 per ROI.

2. Concern is addressed.

3. Concern is addressed.

4. Previous studies were performed on rats. The reviewer supposes the cardiac patch used for pigs would be much larger. Yet, scaling up is not always technically easy. Large tissue patches may suffer from poor perfusion, resulting in a lack of oxygen and nutrients. Therefore, it would still be necessary to characterize the patch even though it is similar to the patches used in prior rat studies.

We thank the reviewer for this important question. We were able to successfully scale up the patch and show viability and function in vitro as well as showing functional benefit in vivo as displayed by the data in this manuscript. Regarding the issue of adequate perfusion in the larger patches, we have already shown that the neonatal fibroblasts on the patch increase blood flow to the damaged myocardium, such that adequate perfusion in the larger patch is not a problem. We added new reference number 36 defining increase blood flow and angiogenesis with the patch (Lancaster et al., (2010). Viable fibroblast matrix patch induces angiogenesis and increases myocardial blood flow in heart failure after myocardial infarction. *Tissue Eng Part A*, 16(10), 3065-3073. <https://doi.org/10.1089/ten.TEA.2009.0589>) This proves the scalability of the patch to this swine size which is the same as what will be used in humans. This information has been added to the text (Page 26; Lines 5-10).

5. Concern is addressed.

Reviewer #2 (Remarks to the Author):

No additional comments. They have made changes according to the reviews.

We thank the reviewer for their positive comments.

Reviewer #3 (Remarks to the Author):

The authors have addressed all my initial concerns, and the manuscript is much improved

We thank the reviewer for their positive comments.

Porcine and Murine Patch Production, Characterization, and Scalability: A proprietary bioresorbable mesh 5cm in diameter with a thickness of 2mm was sterilized using ethylene oxide and allowed to degas for two weeks. Previously expanded neonatal dermal fibroblasts (NDFs) and hiPSC-CMs were thawed and cultured to incorporate them into the sterile matrix disks. Cultures were maintained at 37°C and 5% CO₂ for 30 days. We used hiPSC-CMs that undergo a 20-day differentiation process and express greater than 90% cardiac troponin-T. The human NDFs were obtained from human foreskin and were expanded for patch fabrication. Patches contain a cardiomyocyte-to-fibroblast ratio of 1:1 to 2:1. The seeded matrix disks (patches) were cultured in RPMI and B27 with medium being exchanged every 24-72 hrs. For characterization of function prior to cryopreservation, the patches were visually assessed for the fibroblasts having completely filled the interstices of the mesh as well as synchronous contractions across the patch. At the completion of culture cardiac patches were cryopreserved. Porcine patches were placed in Origen CS50 (Austin, TX, USA) cryopreservation biobags, sealed, filled with Cryostor® CS10 (Biolife Solutions®, Bothell, WA, USA) cryopreservation solution, and subjected to controlled and stepwise cooling process to -90°C. The bags were transferred to <-170°C and stored in vapor phase liquid nitrogen. Murine sized patches were obtained from larger patches maintaining the 1:1 to 2:1 ratio as described above. Prior to implantation, the cardiac patch was thawed using a 37°C water bath and maintained in RPMI with B27 at 37°C and 5% CO₂. Prior to implantation, visual observations of synchronous contraction, integrity of the biomaterial, cell composition, and morphology were used as a characterization to confirm integrity and function. Prior to transfer to the surgical suite, the medium was removed and replaced with 1X Phosphate-Buffered Saline (PBS). An approach in which cell ratios and culture conditions were maintained and matrix was scaled by area allowed us to successfully scale the patch as evidenced by functional benefit *in vivo* as displayed by the data in this manuscript. This text has been added to the manuscript (Page 24; Lines 21-33, Page 25, Lines 1-10).

For better clarity, We have revised Table 1 by moving the heart rate data to be part of the Conductance Catheter Parameters and placed the T2 measurements to be part of the Magnetic Resonance Imaging Parameters. These changes do not affect the results or the conclusions.

In reviewing the revised manuscript, we found an error in legend for Figure 3 and graphical errors in Figures 10, 11 and 12. These errors do not alter the claims made within the text of the results or discussion. For Figure 3 we omitted stating left ventricular volumes within the legend and addressed this by updating the legend to accurately describe the graphs presented. For Figures 10, 11 and 12 we have provided updated figures with the corrected data presented. For transparency to the reviewers, we have also provided the data associated with these figures in a table isolated from the transcriptomic studies completed that support this correction in Supplemental Table 1 for Reviewers.

Figure 3. The legend has been corrected to include both LV and RV volume remodeling and we added brackets to the statistical comparison within groups.

Figure 3. Left ventricular (LV) and right ventricular (RV) volumes: LV and RV volumes at 3 months and 6 months showed similar trends with increases in both at 3 months and 6 months in the control animals, while RV end-diastolic volume decreased both at 3 months and 6 months after patch placement. The data for LV volumes at 3 months was the same as at 6 months. Abbreviations: LV; left ventricle, RV; right ventricle. See new Figure 3 legend on page 9.

Figure 10. The corrected data show increases in CD68 and AFGRE 1 in the Patch CD 45(+) showing an increase in immune cells with the patch. The corrected figure has been added to the revised text on page 14.

Fig. 10. Macrophage abundance CD45^{pos} regions. High macrophage abundance within CD45^{pos} infarct region defined by increases in PTPRC, CD68, and ADGRE expression compared to lower expression in CD45^{neg} non-treated and treated infarct zones. * P<0.05, *P<0.001. Abbreviations: PTPRC: protein tyrosine phosphatase receptor signaling molecules that regulate cell growth, differentiation, mitosis, and oncogenic transformation; CD68: Routinely used as a histochemical/cytochemical marker of inflammation associated with the involvement of monocytes/macrophages; ADGRE: Adhesion G protein coupled receptor, activated in dendritic cell development.

Figure 11 the corrected data shows significance between the CD45- and CD45+ patch treated regions, with significance < 0.05 in the Dendritic Relative Abundance.

Figure 11. Macrophage/Dendritic cells. Relative Abundance (left) and Dendritic Relative Abundance (right) increase in CD45^{pos} treated infarct zones, compared to MI control and CD45^{neg} regions. Abbreviations: MI: Myocardial Infarction.

Figure 12. The corrected data show increases in RETNLA and MRC 1 in the Patch CD45(+) zones confirming the increase in M2 macrophage phenotype with the patch. The corrected figure has been added to the revised text on page 15.

Figure 12. RETNLA and MRC1 Expression. The cardiac patch polarizes macrophages to anti-inflammatory states within the CD45^{pos} regions as defined by increases in RETNLA and MRC1 expression, compared to CD45^{neg} non-treated and treated infarct zones. Abbreviations: RETNLA: Alternatively activated macrophage marker for M2 phenotype; MRC1: Mannose Receptor C-Type 1 is a membrane receptor that mediates the endocytosis of glycoproteins by macrophages.

For transparency to the reviewers, we have also provided the data associated with these figures in a table isolated from the transcriptomic studies completed that support this correction in Supplemental Table 1 for Reviewers.

Supplemental Table 1 for Reviewers.

MI Control									
Sample Name	Group	Region of Interest	Cell Type	Treatment Type	PTPRC	CD68	ADGRE1	MRC1	RETNLA
G2_MI/001/CD45_Neg	G2_MI	Infarct Zone; Non-Treated	CD45_Neg	Non-Treated	22.31	23.68	31.57	45.99	12.35
G2_MI/003/CD45_Neg	G2_MI	Infarct Zone; Non-Treated	CD45_Neg	Non-Treated	29.10	30.72	35.57	39.34	6.47
G2_MI/004/CD45_Neg	G2_MI	Infarct Zone; Non-Treated	CD45_Neg	Non-Treated	10.87	17.40	23.19	23.92	4.35
G2_MI/004/CD45_Neg	G2_MI	Infarct Zone; Non-Treated	CD45_Neg	Non-Treated	40.80	31.53	38.02	43.28	10.82
G2_MI/005/CD45_Neg	G2_MI	Infarct Zone; Non-Treated	CD45_Neg	Non-Treated	22.70	33.21	37.41	34.89	10.93
G2_MI/006/CD45_Neg	G2_MI	Infarct Zone; Non-Treated	CD45_Neg	Non-Treated	28.38	31.53	35.73	36.78	9.46
G2_MI/007/CD45_Neg	G2_MI	Infarct Zone; Non-Treated	CD45_Neg	Non-Treated	25.69	25.69	35.03	26.86	3.50
G2_MI/008/CD45_Neg	G2_MI	Infarct Zone; Non-Treated	CD45_Neg	Non-Treated	15.46	35.86	29.06	24.11	7.42
G2_MI/008/CD45_Neg	G2_MI	Infarct Zone; Non-Treated	CD45_Neg	Non-Treated	45.19	35.73	52.55	44.67	6.83
				AVG	26.72	29.48	35.35	35.54	8.01
				SEM	3.66	2.03	2.66	2.91	1.02
Patch CD45 (-)									
Sample Name	Group	Region of Interest	Cell Type	Treatment Type	PTPRC	CD68	ADGRE1	MRC1	RETNLA
G2_MI-Patch/002/CD45_Neg	G2_MI-Patch	Infarct Zone; Treated	CD45_Neg	Treated	16.52	16.52	36.03	18.02	25.52
G2_MI-Patch/006/CD45_Neg	G2_MI-Patch	Infarct Zone; Treated	CD45_Neg	Treated	25.15	27.40	38.66	31.53	9.01
G2_MI-Patch/009/CD45_Neg	G2_MI-Patch	Infarct Zone; Treated	CD45_Neg	Treated	17.52	17.52	32.41	22.77	8.32
G2_MI-Patch/009/CD45_Neg	G2_MI-Patch	Infarct Zone; Treated	CD45_Neg	Treated	24.34	28.76	35.40	22.13	17.70
G2_MI-Patch/012/CD45_Neg	G2_MI-Patch	Infarct Zone; Treated	CD45_Neg	Treated	23.02	41.04	43.04	37.03	15.01
G2_MI-Patch/012/CD45_Neg	G2_MI-Patch	Infarct Zone; Treated	CD45_Neg	Treated	21.83	23.85	39.21	27.49	15.76
G2_MI-Patch/013/CD45_Neg	G2_MI-Patch	Infarct Zone; Treated	CD45_Neg	Treated	22.13	29.32	43.70	54.76	52.00
G2_MI-Patch/016/CD45_Neg	G2_MI-Patch	Infarct Zone; Treated	CD45_Neg	Treated	22.33	22.33	28.90	35.47	254.86
				AVG	21.60	25.84	37.17	31.15	49.77
				SEM	1.08	2.76	1.79	4.12	29.71
Patch CD45 (+)									
Sample Name	Group	Region of Interest	Cell Type	Treatment Type	PTPRC	CD68	ADGRE1	MRC1	RETNLA
G2_MI-Patch/002/CD45_Pos	G2_MI-Patch	Infarct Zone; Treated	CD45_Pos	Treated	77.77	42.04	37.84	42.04	37.84
G2_MI-Patch/006/CD45_Pos	G2_MI-Patch	Infarct Zone; Treated	CD45_Pos	Treated	70.07	51.38	74.74	44.37	63.06
G2_MI-Patch/009/CD45_Pos	G2_MI-Patch	Infarct Zone; Treated	CD45_Pos	Treated	81.08	51.05	84.08	78.07	21.02
G2_MI-Patch/009/CD45_Pos	G2_MI-Patch	Infarct Zone; Treated	CD45_Pos	Treated	63.06	63.06	56.75	84.08	203.89
G2_MI-Patch/012/CD45_Pos	G2_MI-Patch	Infarct Zone; Treated	CD45_Pos	Treated	73.57	84.08	63.06	52.55	21.02
G2_MI-Patch/013/CD45_Pos	G2_MI-Patch	Infarct Zone; Treated	CD45_Pos	Treated	78.07	33.03	57.05	63.06	330.31
G2_MI-Patch/016/CD45_Pos	G2_MI-Patch	Infarct Zone; Treated	CD45_Pos	Treated	63.06	37.84	54.65	67.26	1147.67
				AVG	72.38	51.78	61.17	61.63	260.69
				SEM	2.75	6.57	5.64	6.12	154.18

Supplemental Table1 for Reviewers: Spatially resolved transcriptomic data of the infarct regions in MI control, Patch treated CD45(-), and Patch treated CD45(+). Individual sample values provided for all macrophage characterization markers: PTPRC(CD45), CD68, ADGRE1, MRC1, and RETNLA.

REVIEWERS' COMMENTS:

Reviewer #1 (Remarks to the Author):

The authors have addressed all my concerns.